# Myoglobin-loaded gadolinium nanotexaphyrins for oxygen synergy and imaging-guided radiosensitization therapy

Xiaotu Ma[1,2,8], Xiaolong Liang[2,8], Meinan Yao[3], Yu Gao[1], Qi Luo[4], Xiaoda Li[5], Yue Yu[1], Yining Sun[1], Miffy H. Y. Cheng [6], Juan Chen[6], Gang Zheng [6,7] ✉, Jiyun Shi [1] ✉ & Fan Wang [1,3,4] ✉

Gadolinium ($Gd^{3+}$)-coordinated texaphyrin (Gd-Tex) is a promising radiosensitizer that entered clinical trials, but temporarily fails largely due to insufficient radiosensitization efficacy. Little attention has been given to using nanovesicles to improve its efficacy. Herein, Gd-Tex is transformed into building blocks "Gd-Tex-lipids" to self-assemble nanovesicles called Gd-nanotexaphyrins (Gd-NTs), realizing high density packing of Gd-Tex in a single nanovesicle and achieving high Gd-Tex accumulation in tumors. To elucidate the impact of $O_2$ concentration on Gd-Tex radiosensitization, myoglobin (Mb) is loaded into Gd-NTs (Mb@Gd-NTs), resulting in efficient relief of tumor hypoxia and significant enhancement of Gd-Tex radiosensitization, eventually inducing the obvious long-term antitumor immune memory to inhibit tumor recurrence. In addition to $Gd^{3+}$, the versatile Mb@Gd-NTs can also chelate $^{177}Lu^{3+}$ (Mb@$^{177}$Lu/Gd-NTs), enabling SPECT/MRI dual-modality imaging for accurately monitoring drug delivery in real-time. This "one-for-all" nanoplatform with the capability of chelating various trivalent metal ions exhibits broad clinical application prospects in imaging-guided radiosensitization therapy.

Radiotherapy (RT) is the most effective cytotoxic therapy available for the treatment of localized solid cancers[1,2]. More than 50% of patients with malignant tumors require RT[1,3]. RT is widely used throughout all cancer treatment stages in the forms of adjuvant RT, neoadjuvant RT, radical RT, and palliative RT[4,5]. The therapeutic efficacy of RT relies on the radiation dose, but increasing the dose will aggravate the radiation damage to normal tissues and organs[6,7]. RT alone also has limited treatment efficacy with a high rate of tumor metastasis and recurrence[7,8]. Therefore, various radiosensitizers have been developed to improve the treatment

efficacy of RT. However, there are few U.S. Food and Drug Administration (FDA) approved radiosensitizers, and the development of radiosensitizers faces some clinical challenges[8,9]. Chemotherapeutics such as cisplatin, paclitaxel, and pentafluorouracil have certain radiosensitization effects, but they are highly toxic in vivo[10]. Hyperbaric oxygen therapy also has some radiosensitization effects, but it highly increases the radiation damage to normal tissues[8]. EGFR inhibitors can also be used for radiosensitization but are only suitable for a small proportion of patients[11]. In addition, radiosensitizers based on inorganic heavy

[1]Key Laboratory of Biomacromolecules, CAS Center for Excellence in Biomacromolecules, Institute of Biophysics, Chinese Academy of Sciences, 100101 Beijing, P. R. China. [2]Department of Ultrasound, Peking University Third Hospital, 100191 Beijing, P. R. China. [3]Medical Isotopes Research Center and Department of Radiation Medicine, State Key Laboratory of Natural and Biomimetic Drugs, School of Basic Medical Sciences, International Cancer Institute, Peking University, 100191 Beijing, P. R. China. [4]Guangzhou National Laboratory, 510005 Guangzhou, P.R. China. [5]Medical and Health Analysis Center, Peking University, 100191 Beijing, P. R. China. [6]Princess Margaret Cancer Centre, University Health Network, Tronto, ON M5G 1L7, Canada. [7]Department of Medical Biophysics, University of Toronto, Tronto, ON M5G 1L7, Canada. [8]These authors contributed equally: Xiaotu Ma, Xiaolong Liang. ✉e-mail: gang.zheng@uhnres.utoronto.ca; shijiyun@ibp.ac.cn; wangfan@bjmu.edu.cn

metal nanomaterials, such as nanogold and nanotitanium, are difficult to degrade and have long-term safety hazards[12,13]. Therefore, the development of nontoxic, highly effective radiosensitizers remains a formidable challenge. Immunotherapy is an emerging and promising way to enhance the efficacy of radiotherapy[14]. Nanomedicine with excellent biodegradability and biocompatibility provides opportunities to facilitate the combination of immunotherapy and radiotherapy[15].

Gadolinium ion ($Gd^{3+}$)-coordinated texaphyrin (Gd-Tex) was one of the most promising radiosensitizers developed by the J.L. Sessler research group[16]. The corresponding commercial drug Motexafin gadolinium (Xcytrin®) developed by Pharmacyclics® in the United States had undergone clinical trials[17,18]. Although Gd-Tex was eventually not approved by the US FDA due to limited radiosensitization efficacy, the accomplishment of its phase III clinical trials proved that it had good safety. If the radiosensitization efficacy of Gd-Tex could be effectively improved, it is hopeful of achieving its successful clinical application, benefiting a large number of patients receiving radiotherapy.

Gd-Tex has a short half-life in blood and limited accumulation in tumors, leading to its limited radiosensitization efficacy[19]. Little attention has been given to using nanovesicles to improve the radiosensitization efficacy of Gd-Tex. Herein, as illustrated in Fig. 1a, we transformed Gd-Tex into building blocks "Gd-Tex-lipids" to self-assemble the liposome-like nanovesicles with lipid bilayer called Gd-nanotexaphyrins (Gd-NTs). Gd-NTs theoretically contains as many as 80,000 Gd-Tex-lipids building blocks in each nanovesicle[20,21]. Therefore, tremendous Gd-Tex could flood into tumors after Gd-NTs passively targeted to tumors via the enhanced permeability and retention (EPR) effect. Besides, the polyethylene glycol (PEG) shell of Gd-NTs largely prolonged the half-life of Gd-Tex. Eventually, the tumor accumulation of Gd-Tex could be largely increased, leading to its enhanced radiosensitization efficacy (as illustrated in Fig. 1b).

The radiosensitization efficacy of Gd-Tex was further increased by the in-depth study of its radiosensitization mechanism. Gd-Tex achieves radiosensitization effects through the speculated theory of "futile redox cycling"[18,22]. Although $O_2$ is proposed to be involved in this theory, it has not been confirmed whether $O_2$ concentration could influence its radiosensitization efficacy. Hypoxia is the hallmark of most solid tumors[23]. It is essential to elucidate the $O_2$ dependence of Gd-Tex radiosensitization. Therefore, the oxygen-carrying protein myoglobin (Mb) was loaded into Gd-NTs (Mb@Gd-NTs), which enabled spatiotemporal codelivery of $O_2$ and high density of Gd-Tex into tumors, resulting in efficient relief of tumor hypoxia and the significant enhancement of Gd-Tex radiosensitization. Eventually, a synergistic effect by appropriate $O_2$ delivery and enhanced radiosensitization of Gd-Tex has been demonstrated.

Imaging monitoring of drug behaviors in vivo facilitates evaluating and optimizing its therapeutic efficacy[24]. The chelated $Gd^{3+}$ in Mb@Gd-NTs enabled magnetic resonance imaging (MRI)[25]. In addition to $Gd^{3+}$, the versatile nanoplatform Mb@Gd-NTs could also chelate $^{177}Lu^{3+}$ (Mb@$^{177}Lu$/Gd-NTs), enabling single-photon emission computed tomography (SPECT) imaging (as illustrated in Fig. 1b). The dual-modality imaging combined the advantages of both modalities, accurately achieving non-invasive and whole-body tracking of drug delivery. The real-time imaging assessment of drug concentration in tumors facilitated selecting the optimal time of radiotherapy.

In this work, the radiosensitization efficacy of Gd-Tex is effectively improved by optimizing its dosage form and in-depth study of its oxygen-dependence. The synergistic mechanism between appropriate $O_2$ delivery and enhanced Gd-Tex radiosensitization has been demonstrated. This "one-for-all" Gd-Tex nanoplatform demonstrates high biocompatibility, simple construction, capability of chelating various trivalent metal ions, and exhibits broad clinical application prospects in imaging-guided radiosensitization therapy.

## Results

### Fabrication and characterization of Mb@Gd-NTs

Gd-Tex-lipid (Gd-Tex-lipid) was successfully synthesized and verified by ESI-Q-TOF mass spectrometry (Supplementary Fig. 1). NTs and Gd-NTs were fabricated by the self-assembly of Tex-lipid and Gd-Tex-lipid using a facile filming-rehydration method. Hemoglobin (Hb)- or myoglobin (Mb)-loaded Gd-NTs (Hb@Gd-NTs or Mb@Gd-NTs) were fabricated using the same method (as shown in the illustration of Fig. 1a). The unencapsulated Mb could be almost completely removed by dialysis, which was confirmed by size exclusion chromatography (Supplementary Fig. 2). All four kinds of NTs had a spherical morphology characterized by transmission electron microscopy (TEM) (Fig. 1c) with a uniform hydrodynamic diameter of ~120 nm (Fig. 1d). UV–Vis absorption spectra confirmed the successful coordination of $Gd^{3+}$ with Tex-lipid and the high density of Gd-Tex-lipid in Gd-NTs (Fig. 1e). Both Gd-NTs and Mb@Gd-NTs had good stability, evidenced by minimal change in particles' hydrodynamic diameter, polydispersity index and zeta potential after 14 days storage at 4 °C (Supplementary Fig. 3) or after three freeze–thaw cycles (Supplementary Fig. 4). The NTs, Gd-NTs, and Mb@Gd-NTs demonstrated good biocompatibility with negligible cytotoxicity to various normal human cell lines (Supplementary Fig. 5).

To examine oxygen delivery efficiency, oxygen-saturated Mb@NTs (MbO_2@NTs) were injected into oxygen-free PBS, and the released oxygen was measured using a dissolved oxygen meter. Mb@NTs could effectively absorb and gradually release oxygen (Fig. 1f). The oxygen delivery efficiency of Mb@NTs was similar to that of free Mb (Fig. 1g), demonstrating that the fabrication procedures of Mb@NTs did not harm the bioactivity of Mb. The oxygen release profile of MbO_2@NTs was quite different from that of HbO_2@NTs. MbO_2@NTs showed a steady release of oxygen, while HbO_2@NTs rapidly released oxygen (Fig. 1g). This difference possibly resulted from the different affinities of the two proteins for oxygen. Hb is composed of four subunits, and there is an allosteric effect between the subunits, while Mb has only one subunit. Under hypoxic conditions, the allosteric effect causes Hb to have a lower affinity for oxygen than Mb, resulting in rapid oxygen releasing[26].

The mechanism of the radiosensitization effect of Gd-Tex is complex and multifaceted[18]. The "futile redox cycling" theory is one of the speculated mechanisms: electrophilic Gd-Tex can easily oxidize cellular reducing metabolites and turn itself into its free radical form (Gd-Tex·), which further reacts with oxygen to generate reactive oxygen species (ROS)[22]. Therefore, radiosensitization is achieved by depleting reducing metabolites and producing ROS. The ability of Mb@Gd-NTs to oxidize ascorbate was measured to simulate the in vivo reaction between Mb@Gd-NTs and reducing substances. Both NTs ($Gd^{3+}$-free) and MbO_2@NTs barely oxidized ascorbate, while Gd-NTs effectively oxidized ascorbate (Fig. 1h), confirming the previous finding that Tex had minimal radiosensitization effect when lacking of $Gd^{3+}$ coordination. In addition, MbO_2@Gd-NTs demonstrated significantly higher and faster oxidation of ascorbate than Gd-NTs (Fig. 1h), indicating that the efficient delivery of oxygen by myoglobin-loaded particles facilitated the oxidation reaction.

### In vitro radiosensitization effect of MbO_2@Gd-NTs and the synergistic effect between Gd-Tex and oxygen

We first investigated if Mb/Hb-loaded nanotexaphyrins could relieve hypoxia of tumor cells. Lewis' lung cancer (LLC) cells were maintained at 1% $O_2$ to induce hypoxic status, followed by incubation with Mb/Hb-loaded nanotexaphyrins. Both NTs and Gd-NTs without carrying Mb/Hb barely influenced the hypoxia status (Fig. 2a, Supplementary Fig. 6a). In contrast, free MbO_2, MbO_2@NTs and MbO_2@Gd-NTs effectively and equally relieved cellular hypoxia, demonstrating that Mb-loaded nanotexaphyrins could deliver $O_2$ to hypoxic cells and that the lipid bilayer did not hamper the free diffusion of $O_2$. Compared

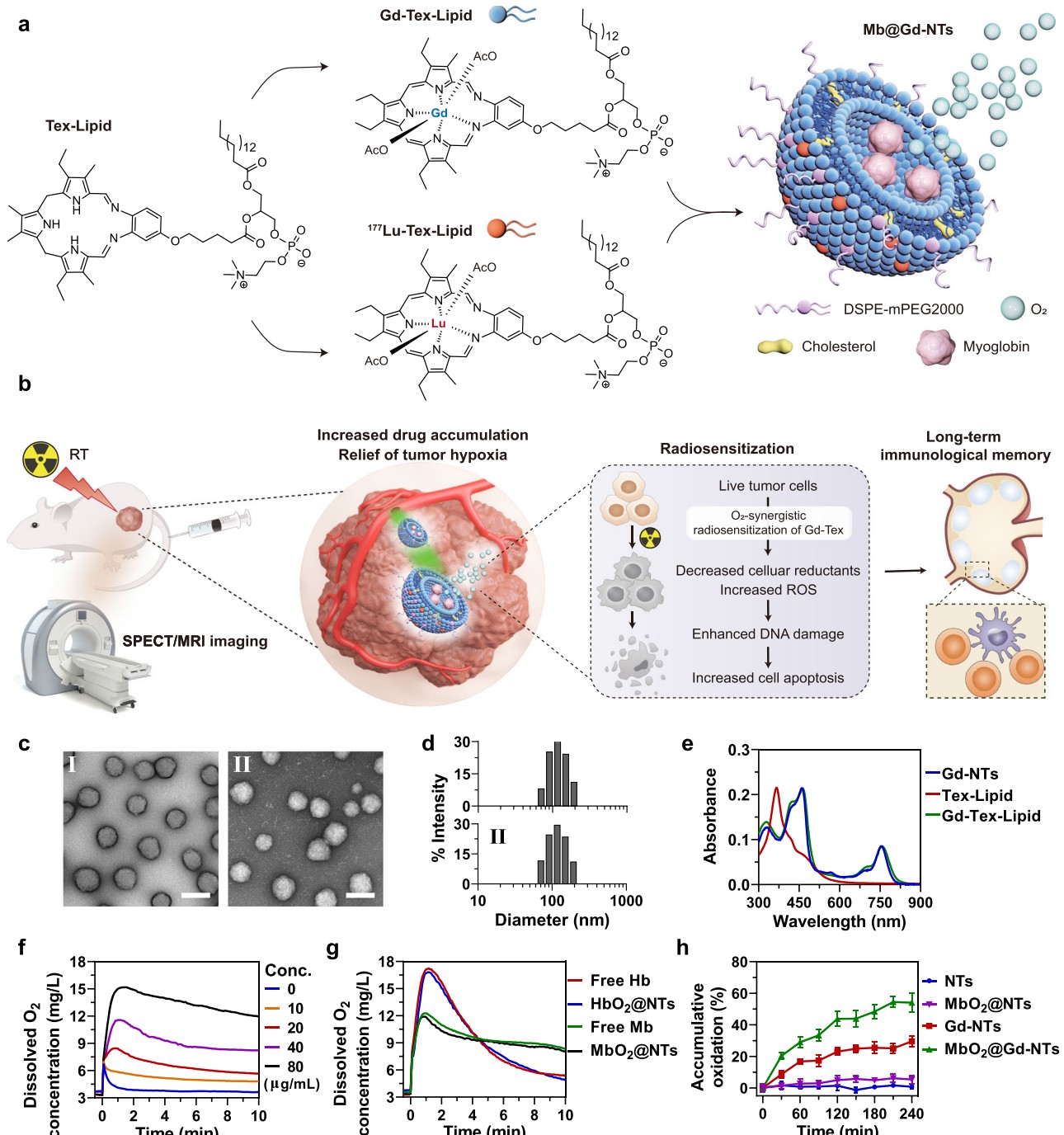

**Fig. 1 | Fabrication and characterization of Mb@¹⁷⁷Lu/Gd-NTs. a** Diagram of the fabrication of Mb@¹⁷⁷Lu/Gd-NTs. **b** Illustration of imaging-guided radio-sensitization therapy. Intravenous injection of Mb@¹⁷⁷Lu/Gd-NTs enables SPECT/MRI dual-modality imaging for accurately monitoring drug delivery in real-time. Mb@¹⁷⁷Lu/Gd-NTs can significantly increase Gd-Tex accumulation in tumors and obviously relieve tumor hypoxia. $O_2$ synergically enhances the radiosensitization effect of Gd-Tex, leading to the decreased reductants and increased ROS in tumor cells, causing the enhanced cell apoptosis, eventually inducing long-term immunological antitumor memory. **c, d** TEM images (**c**) and hydrodynamic diameter distribution (**d**) of Gd-NTs (I) and Mb@Gd-NTs (II). Scale bar, 100 nm. These experiments (**c, d**) were repeated three times independently with similar results.

**e** The UV–Vis absorption spectra of Tex-lipid, Gd-Tex-lipid and Gd-NTs. The experiment (**e**) was repeated three times independently with similar results. **f** The concentration of dissolved oxygen released from different concentrations of $MbO_2$@Gd-NTs (0-80 μg/mL Mb). **g** The concentration of dissolved oxygen released from free Hb, $HbO_2$@NTs, free Mb, and $MbO_2$@NTs (40 μg/mL Hb or Mb). $MbO_2$, oxygenated Mb. $HbO_2$, oxygenated Hb. These experiments (**f, g**) were repeated three times independently with similar results. **h** The kinetics of ascorbate oxidation catalyzed by NTs, $MbO_2$@NTs, Gd-NTs, $MbO_2$@Gd-NTs under hypoxic conditions. The data (**h**) are shown as the mean ± SD ($n = 3$ independent experiments). Source data are provided as a Source Data file.

with free $HbO_2$ and $HbO_2$@NTs, free $MbO_2$ and $MbO_2$@NTs alleviated hypoxia better, probably because $MbO_2$ released $O_2$ steadily (Fig. 1g).

The radiosensitization effect in vitro and the underlying mechanism were then investigated. Upon X-ray irradiation with a dose of 2 Gy, unlike NTs (Gd³⁺-free) showed nonsignificant ROS generation, Gd-NTs demonstrated effective RT response with significant ROS increase in LLC cells (Fig. 2b, c) due to the deprivation of reducing substances and the generation of superoxide anions. The increased

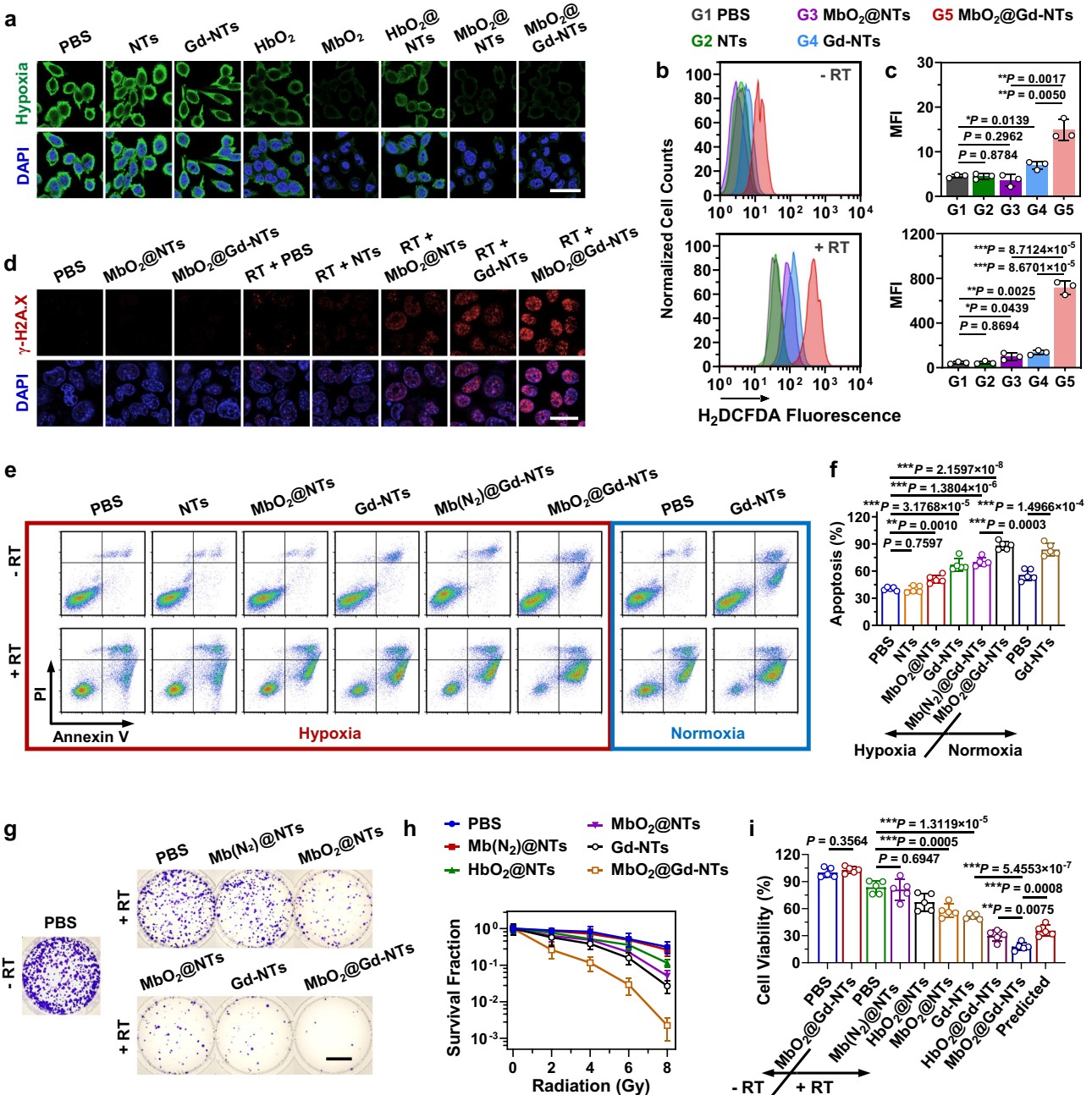

**Fig. 2 | In vitro radiosensitization effect of MbO₂@Gd-NTs and the synergistic effect between Gd-Tex and O₂. a** Hypoxia level of LLC cells after incubating cells with different agents under hypoxic or normoxic conditions. Each group of these experiments (**a**) was performed on 4 independent samples with similar results (c.f. Supplementary Fig. 6a). Scale bar, 20 μm. **b, c** Flow cytometry analysis of cellular ROS levels before and after radiotherapy (RT) using X-ray irradiation (total 2 Gy for one time, *n* = 3 independent samples). The ROS levels were indicated by H2DCFDA fluorescence. **d** Cellular DNA DSBs examined by the immunofluorescence staining of γ-H2A.X after incubating cells with different agents under hypoxic conditions and treating cells using X-ray irradiation (total 2 Gy for one time). Scale bar, 40 μm. Each group of these experiments (**d**) was performed on 5 independent samples with similar results (c.f. Supplementary Fig. 6b). **e, f** Cellular apoptosis was examined by Annexin V-FITC/PI double staining after incubating cells with different agents under

hypoxic or normoxic conditions and treating cells using X-ray irradiation (total 2 Gy for one time). Quantification results of cellular apoptosis with X-ray irradiation were shown (**f**, *n* = 5 independent samples). **g, h** In vitro radiosensitization effect examined by colony formation assay. Photographs of colonies (**g**) and survival fraction (**h**) of LLC cells. Cells were treated with different agents under hypoxic conditions, and irradiated with X-rays at total doses of 2, 4, 6, or 8 Gy for one time (*n* = 4 independent samples). Scale bar, 2 cm. **i** Cell viability examined by CCK-8 assay. Cells were treated with different agents under hypoxic conditions before X-ray irradiation (total 2 Gy for one time, *n* = 5 independent samples). The data (**c, f, h, i**) are shown as the mean ± SD. Statistical analysis was performed by a two-tailed unpaired *t* test (**c, f, i**). *P < 0.05; **P < 0.01; ***P < 0.001. Source data are provided as a Source Data file.

ROS level by Gd-NTs led to a higher degree of DNA double strand breaks (DSBs) (Fig. 2d, Supplementary Fig. 6b) and cellular apoptosis (Fig. 2e, f, Supplementary Fig. 7), and eventually caused an obviously reduced clonogenic survival fraction (Fig. 2g, h) and cellular metabolic activity (Fig. 2i) compared with PBS + RT. Therefore, Gd-NTs showed a

significant radiosensitization effect, while NTs did not. MbO₂@NTs without Gd chelation also exhibited an obvious radiosensitization effect, and the clonogenic survival fraction was reduced by 83.9% compared with PBS + RT (8 Gy, Fig. 2h), which was probably contributed by the delivery of O₂. Importantly, MbO₂@Gd-NTs + RT

induced the most significant elevation of ROS, DSB, and apoptosis levels, which were respectively increased by 15.4 folds (Fig. 2c), 11.8 folds (Supplementary Fig. 6b), 1.2 folds (Fig. 2f) compared with PBS + RT (2 Gy). Consequently, MbO$_2$@Gd-NTs + RT caused the highest reduction of clonogenic survival fraction by 99.3% compared with PBS + RT (8 Gy, Fig. 2h), and the highest reduction of cellular metabolic activity by 78.6% compared with PBS + RT (2 Gy, Fig. 2i). Moreover, MbO$_2$@Gd-NTs demonstrated a superior RT effect over the calculated predicted additive effect of Gd-NTs and MbO$_2$@NTs (MbO$_2$@Gd-NTs > Gd-NTs + MbO$_2$@NTs, "1 + 1 > 2") with significant reduced cellular metabolic activity (Fig. 2i), indicating that there was an obvious synergistic effect between Gd-NTs and MbO$_2$.

## Single-photon emission computed tomography (SPECT) and magnetic resonance imaging (MRI) bimodal imaging, pharmacokinetics and biodistribution studies

As high-density packing of Gd-Tex-lipid in a single Mb@Gd-NT, we investigate its potential as an MRI contrast agent. The in vitro MRI contrast effect of Mb@Gd-NTs was examined at a field strength of 7 T (Fig. 3a), and the longitudinal relaxivity ($r_1$) was calculated by plotting the inverse relaxation time against the Gd$^{3+}$ concentration (Fig. 3b). The intact Mb@Gd-NTs could significantly enhance the MRI signal with an $r_1$ value of 0.62 mM$^{-1}$S$^{-1}$. Surfactant-dissociated Mb@Gd-NTs had a higher r$_1$ value (0.77 mM$^{-1}$S$^{-1}$) than the intact. Therefore, Mb@Gd-NTs might have a better MRI contrast effect after endocytosis and dissociation in tumor cells. Upon intravenous (i.v.) administration, Mb@Gd-NTs largely increased the MRI signal of the tumor region (Fig. 3c). The highest signal was achieved and spread in the whole tumor region at 24 h after injection, demonstrating the highest Mb@Gd-NTs accumulation in the tumor.

As texaphyrin 5-coordination pocket enables stable chelation of large ionic radii of lanthanides[27], we radiolabeled Mb@Gd-NTs with $^{177}$Lu for single photon-emission computed tomography (SPECT). Tex-lipid was first labeled with $^{177}$Lu$^{3+}$ ($^{177}$Lu-Tex-lipid), and the $^{177}$Lu$^{3+}$ labeling procedure could be quickly completed within 30 min. The $^{177}$Lu$^{3+}$-labeled Mb@Gd-NTs (Mb@$^{177}$Lu/Gd-NTs) were obtained by mixing $^{177}$Lu-Tex-lipid and Gd-Tex-lipid to construct the nanoassembly, as illustrated in Fig. 1a. The radiochemical purity of Mb@$^{177}$Lu/Gd-NTs was >98% after purification (Supplementary Fig. 8). The SPECT images of LLC tumor-bearing mice were acquired at 2-72 h after i.v. injection of Mb@$^{177}$Lu/Gd-NTs. The Mb@$^{177}$Lu/Gd-NTs showed clear tumor imaging with high contrast to the contralateral background (Fig. 3d). Tumors achieved the highest uptake of Mb@$^{177}$Lu/Gd-NTs at 24 h post-injection, in accordance with the imaging results of MRI. Biodistribution of Mb@$^{177}$Lu/Gd-NTs at 6–72 h post-injection showed that, except for obvious uptake in tumors, Mb@$^{177}$Lu/Gd-NTs accumulated in the liver and spleen, showing the in vivo behavior characteristics of nanoparticles (Fig. 3e). Among all tested time points, the tumor uptake of Mb@$^{177}$Lu/Gd-NTs was highest at 24 h (6.92 %ID/g).

To track Mb delivery, Cy5.5-labeled Mb (Cy5.5-Mb) was loaded in Mb@Gd-NTs and monitored by fluorescence imaging on LLC tumor-bearing mice. Free Cy5.5-Mb achieved the highest fluorescence signal in tumors at 2 h after injection, but Cy5.5-Mb@Gd-NTs achieved the highest fluorescence signal in tumors at 24 h after injection (Fig. 3f, Supplementary Fig. 9), corresponding to MRI and SPECT imaging in vivo. Therefore, MRI/SPECT imaging could be used for imaging-guided dosimetry to guide treatment plans.

After the homogenization and lysis of tumors and normal organs, Mb concentrations were precisely quantified by fluorescence spectrophotometry (Fig. 3g). The tumor uptake of Mb after injection of Mb@Gd-NTs was 7.88 folds higher than that injection of free Mb (4.62 vs. 0.52 %ID/g measured at 24 h and 2 h after injection, respectively) (Fig. 3g), demonstrating enhanced delivery by Mb@Gd-NTs, which was mainly due to the obviously prolonged blood circulation of Mb@Gd-NTs proven by pharmacokinetics studies (Fig. 3h). The half-lives of the

distribution phase ($t_{1/2\alpha}$) and elimination phase ($t_{1/2\beta}$) for Mb@Gd-NTs were 6.83 times (0.82 h vs. 0.12 h) and 8.10 times (23.98 h vs. 2.96 h) as long as that of free Mb (Supplementary Table 1). In addition, Hb@NTs, Mb@NTs, and Mb@Gd-NTs were quite similar in pharmacokinetics and biodistribution behaviors (Fig. 3f–h) due to the same shell of polyethylene glycol.

Theoretically, the injected Mb@Gd-NTs can absorb O$_2$ in the pulmonary capillaries and convert it to MbO$_2$@Gd-NTs, then release O$_2$ in the tumor hypoxic environment. Therefore, we investigated whether Mb@Gd-NTs could alleviate tumor hypoxia in vivo. As shown in Fig. 3i, j, both Gd-NTs and free Mb could barely influence tumor hypoxia status, while Mb@Gd-NTs significantly relieved tumor hypoxia at 24 h post-injection, which resulted from the largely enhanced tumor uptake (Fig. 3g) and obviously prolonged blood circulation of Mb@Gd-NTs proven by pharmacokinetics studies (Fig. 3h). Mb@Gd-NTs also demonstrated better efficacy in relieving hypoxia than Hb@Gd-NTs (Fig. 3i, j), suggesting the advantage of Mb over Hb in O$_2$ delivery.

## Radiosensitization efficacy of Mb@Gd-NTs in vivo

The in vivo radiosensitization efficacy was assessed in LLC-bearing mice. X-ray radiation at a dose of 2 Gy was given every two days for a total of five times, and different agents were i.v. injected before each radiation (as shown in the schedule of Fig. 4a). Mb@Gd-NTs without radiation had little influence on tumor growth (Supplementary Fig. 10), showing good biocompatibility. Both Gd-NTs and Mb@NTs enhanced the therapeutic efficacy of RT, as the tumor growth of these two groups was delayed (Fig. 4b, c), and the overall survival was improved (Fig. 4d). Importantly, Mb@Gd-NTs achieved more significant enhancement of RT efficacy, and the tumor growth inhibition was much stronger than the calculated predicted additive effect of Gd-NTs and Mb@NTs (Fig. 4b, c), i.e., the radiosensitization efficacy of Mb@Gd-NTs > Gd-NTs + Mb@NTs, "1 + 1 > 2", indicating a synergistic effect caused by Gd-NTs and Mb based O$_2$ delivery. Hematoxylin and eosin (H&E) staining (Fig. 4e), TUNEL (terminal-deoxynucleotidyl transferase mediated nick end labeling) assay (Fig. 4f, Supplementary Fig. 11a), and immunohistochemical staining of γ-H2A.X (Fig. 4g, Supplementary Fig. 11b) of tumor tissue sections further supported the enhanced efficacy as more severe levels of necrosis, DNA single-strand break (SSBs), and DSBs were found in Mb@Gd-NTs + RT group compared with Gd-NTs + RT group or Mb@NTs + RT group.

The O$_2$-dependent radiosensitization of Mb@Gd-NTs had been confirmed by the above in vitro and in vivo assays. The radiosensitization mechanism of Mb@Gd-NTs ("futile redox cycling") was summarized in the illustration of Fig. 4h. RT can generate ROS by the radiolysis of water (pathway 1 in the illustration of Fig. 4h), leading to the temporary damage of DNA (pathway 2), which can be repaired by reductants (pathway 3). Electrophilic Gd-Tex can oxidize reductants and turn itself into its free radical (Gd-Tex•), which reacts with O$_2$ to generate ROS, regenerating Gd-Tex. The decrease of reductants hampers the ability of tumor cells to scavenge ROS (pathway 4) and repair DNA damage (pathway 5), while the increased ROS generated by Gd-Tex can aggregate DNA damage (pathway 6), leading to the fixed DNA damage and cell death (pathway 7). Gd-Tex can also damage other biomacromolecules such as proteins and lipids by similar mechanisms (pathway 8 and 9).

## Comparing radiosensitization efficacy of Mb@Gd-NTs with nimorazole

Nimorazole is a water-soluble 5-nitroimidazole compound that has been widely used in clinic for the RT of head and neck cancers in Denmark since 1990[28,29]. We have compared the radiosensitization efficacy and side effects of Mb@Gd-NTs with the clinically used nimorazole. The radiosensitization mechanism of nimorazole was illustrated in Fig. 5a using hydroxyl radical (•OH) as an example. The

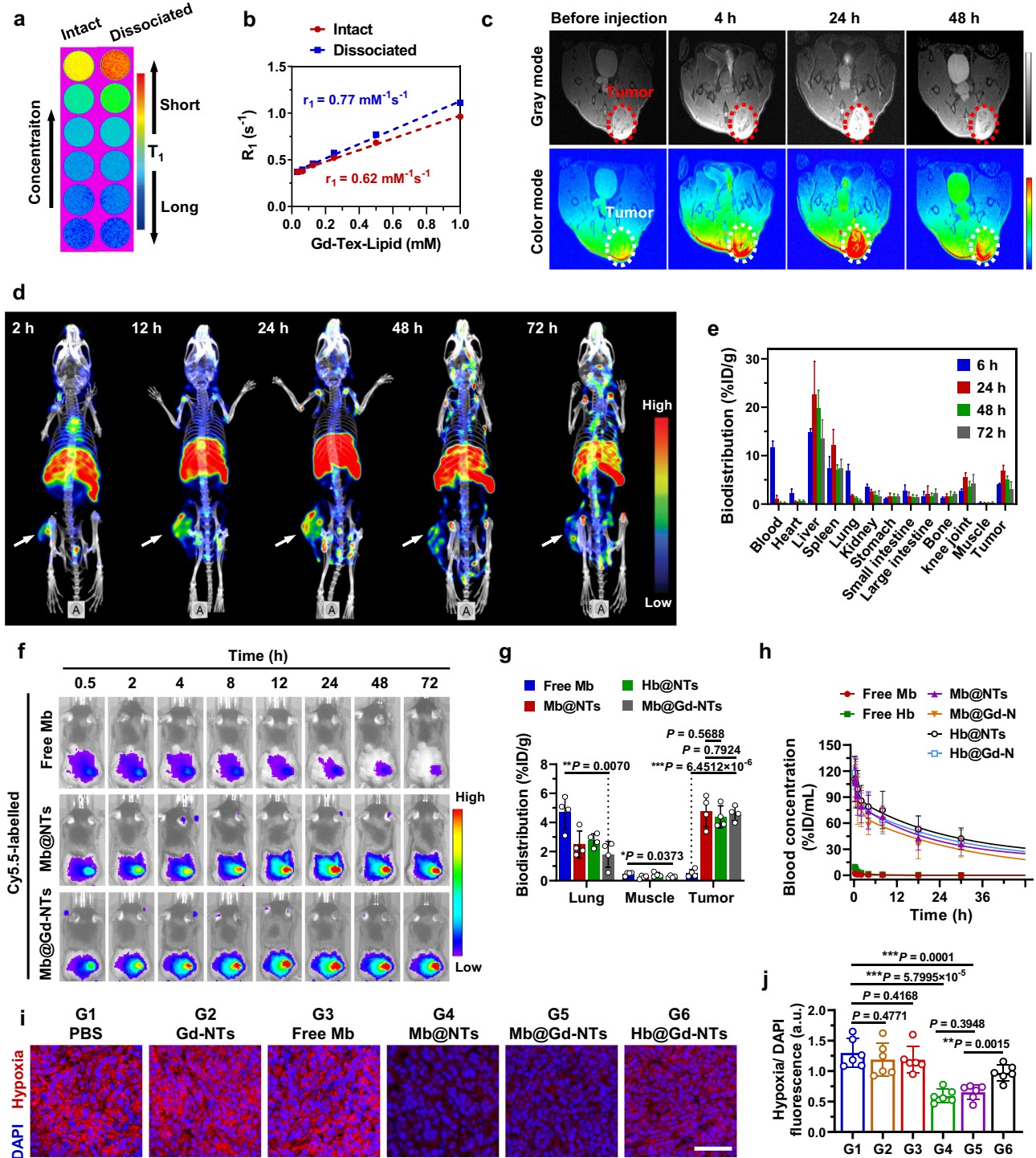

**Fig. 3 | MRI/SPECT/CT imaging, pharmacokinetics and biodistribution of Mb@Gd-NTs. a** In vitro phantom MRI contrast images of different concentrations of Mb@Gd-NTs. **b** Linear fitting of $R_1$ ($1/T_1$) values as a function of different concentrations of Gd-Tex-lipid and the $r_1$ value of the intact or dissociated Mb@Gd-NTs. **c** In vivo $T_1$-weighted MRI imaging of LLC tumor-bearing mice before and 4 h, 24 h, and 48 h after the injection of Mb@Gd-NTs. **d** SPECT/CT images of LLC tumor-bearing mice acquired at 2–72 h after i.v. injection of Mb@$^{177}$Lu/Gd-NTs. These experiments (**a–d**) were repeated three times independently with similar results. **e** Biodistribution of Mb@$^{177}$Lu/Gd-NTs in tumors and major organs measured at 6, 24, 48, and 72 h post-injection. The data (**e**) are shown as the mean ± SD ($n = 4$ mice). **f** In vivo fluorescence imaging of LLC tumor-bearing mice 0.5–72 h after i.v. injection of free Cy5.5-Mb, Cy5.5-Mb@NTs and Cy5.5-Mb@Gd-NTs ($n = 4$ mice, cf.

Supplementary Fig. 9). **g** The content of Cy5.5- Mb or Cy5.5-Hb in lungs, muscles, and tumors 2 h after the injection of free Cy5.5-Mb and 24 h after the injection of Cy5.5-Mb@NTs, Cy5.5-Hb@NTs, and Cy5.5-Mb@Gd-NTs. The data (**g**) are shown as the mean ± SD ($n = 4$ mice). **h** Pharmacokinetic studies of Cy5.5-labeled different agents. The data (**h**) are shown as the mean ± SD ($n = 4$ mice). **i, j** Immunofluorescent staining of the hypoxic region (pseudo-red color) of tumor tissue sections and the corresponding semi-quantification results (**j**). Cell nuclei were stained with DAPI (pseudo-blue color). Tumors were excised from mice 2 h after the injection of free Mb and 24 h after the injection of different liposomes. Scale bar, 60 μm. The data (**j**) are shown as the mean ± SD ($n = 6$ mice). Statistical analysis was performed by a two-tailed unpaired $t$ test (**g, h, j**). $^*P < 0.05$; $^{**}P < 0.01$; $^{***}P < 0.001$. Source data are provided as a Source Data file.

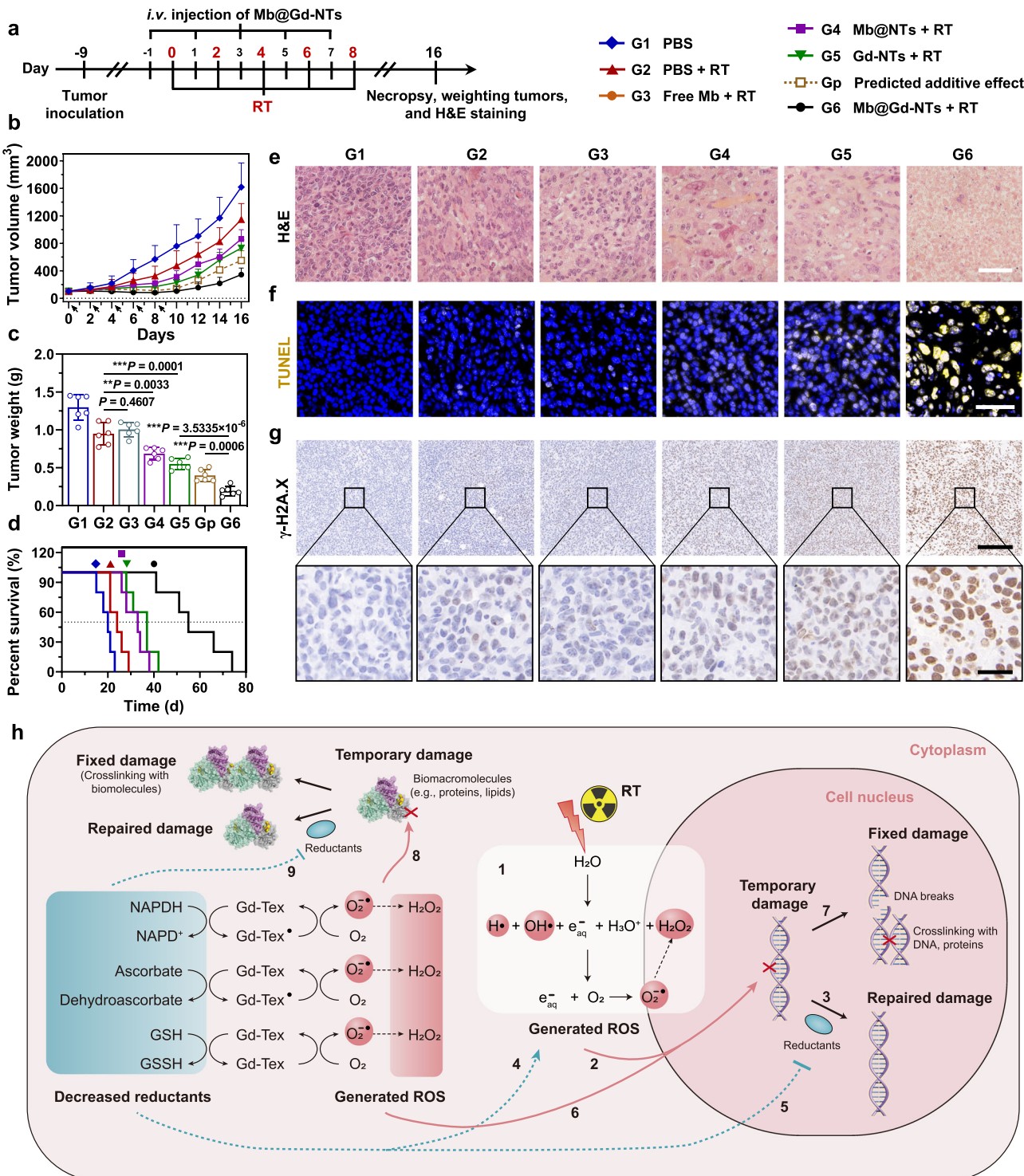

**Fig. 4 | In vivo radiosensitization effect of Mb@Gd-NTs. a** Scheme and grouping of in vivo therapy. The C57BL/6 mice were inoculated with LLC cells on day −9, allowing a tumor volume of -100 mm³ on day 0. The tumor sites were irradiated with a dose of 2 Gy on days 0, 2, 4, 6, and 8 for a total dose of 10 Gy. Free Mb was intravenously injected 2 h before each RT session, whereas texaphyrin-contained agents were intravenously injected 24 h before each RT session with a texaphyrin dose of 5 µmol/kg (6.79 mg/kg). **b** The change in tumor volume after five rounds of RT (indicated by the black arrows). The data are shown as the mean ± SD (n = 5 mice). **c** The tumor weight of different treatment groups weighed on days 16. The data are shown as the mean ± SD (n = 6 mice). Statistical analysis was performed by a two-tailed unpaired t test. *P < 0.05; **P < 0.01; ***P < 0.001. **d** Survival curves of different treatment groups (n = 5 mice). **e, f** H&E staining (**e**) and TUNEL assay (**f**) of tumor tissue sections. Scale bar: H&E staining, 100 µm; TUNEL assay,

60 µm. **g** γ-H2A.X immunohistochemical staining of tumor tissue sections. Scale bar: 180 µm (upper panel) and 30 µm (bottom panel). These experiments (**e**–**g**) were repeated three times independently with similar results. **h** The summarized illustration of the radiosensitization mechanism of Mb@Gd-NTs by "futile redox cycling". RT can generate ROS by the radiolysis of water (1), leading to the temporary damage of DNA (2), which can be repaired by reductants (3). Gd-Tex can continuously oxidize reductants and generate ROS. The decrease of reductants hampers the ability of tumor cells to scavenge ROS (4) and repair DNA damage (5), while the increased ROS generated by Gd-Tex can aggregate DNA damage (6), leading to the fixed DNA damage and cell death (7). Gd-Tex can also damage other biomacromolecules such as proteins and lipids by similar mechanisms (8, 9). Source data are provided as a Source Data file.

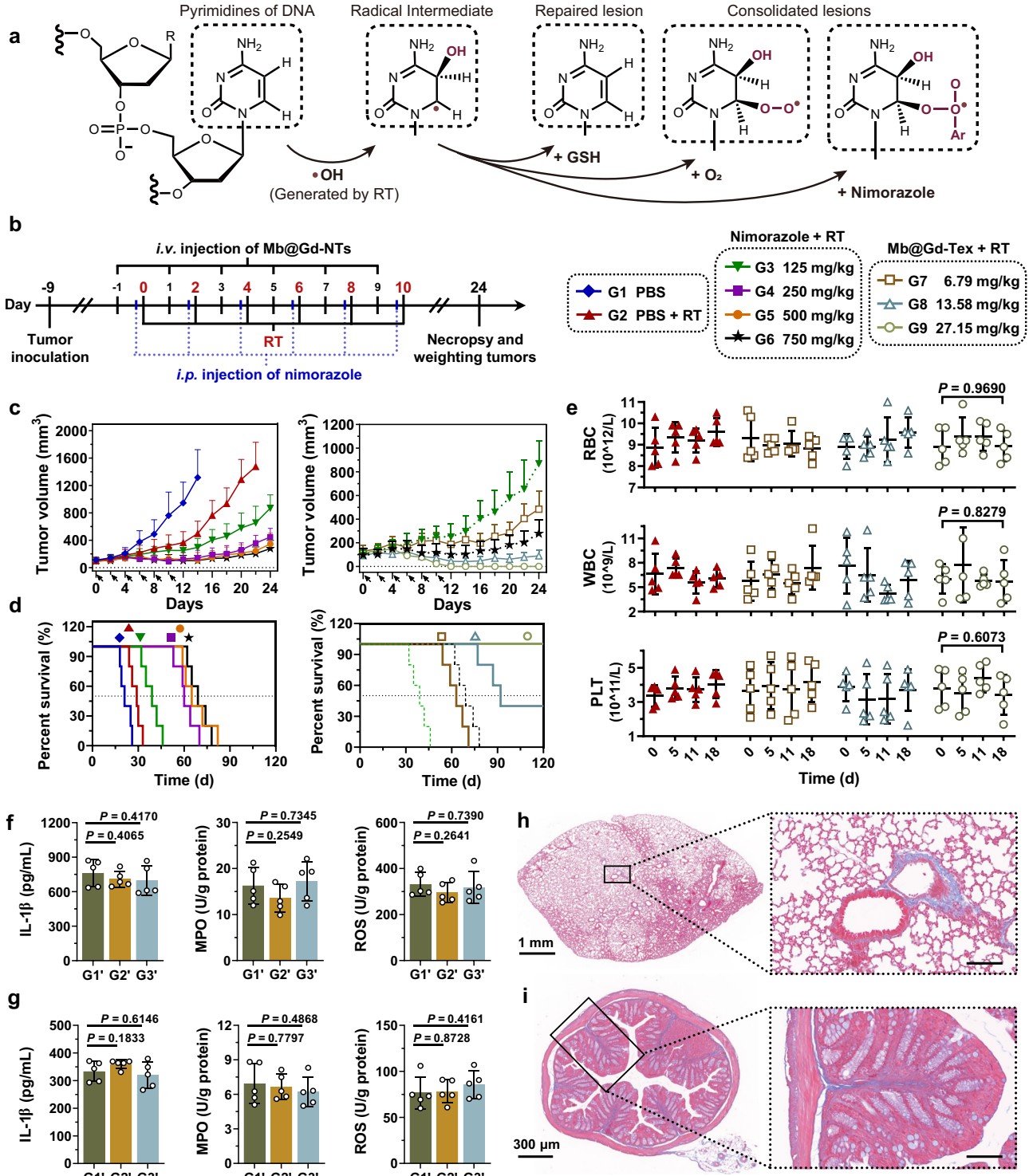

**Fig. 5 | Comparison of radiosensitization effect between Mb@Gd-NTs and clinically used nimorazole. a** Illustration of the radiosensitization mechanism of nimorazole (Ar−NO₂) using hydroxyl radical (•OH) as an example. **b** Scheme and grouping of in vivo therapy. The C57BL/6 mice were inoculated with LLC cells on day −9. The tumor sites were irradiated with a dose of 2 Gy on days 0, 2, 4, 6, 8, and 10 for a total dose of 12 Gy. Mb@Gd-NTs were intravenously injected with different doses of 6.79−27.15 mg/kg 24 h before each RT session, whereas nimorazole was i.p. injected with different doses of 125−750 mg/kg 30 min before each RT session. **c, d** The change in tumor volume (**c**) and the survival curves (**d**) after six rounds of RT indicated by the black arrows. The data are shown as the mean ± SD (*n* = 5 mice). **e** The change in routine blood examination after treating mice with Mb@Gd-NTs + RT (G9, 27.15 mg/kg). The data are shown as the mean ± SD

(*n* = 5 mice). **f–i** The examination of radiation-induced side effects of intestines and lungs. C57BL/6 mice were treated with PBS (G1′), PBS + RT (G2′), and Mb@Gd-NTs + RT (G3′) as shown in the schedule of Supplementary Fig. 16a, whereas mouse abdomen or chest was irradiated instead of tumors with a dose of 2 Gy on days 0, 2, 4, 6, 8, 10 for a total dose of 12 Gy. The oxidation stress and inflammatory responses of the lung (**f**) and intestine (**g**) were analyzed on days 25. Masson staining of lung sections (**h**) and colorectum sections (**i**) showed the distribution of collagen (blue) in tissues. Scale bar of the right panels, 100 µm. The data (**f**, **g**) are shown as the mean ± SD (*n* = 5 mice). Statistical analysis was performed by a two-tailed unpaired *t* test. *P < 0.05; **P < 0.01; ***P < 0.001. These experiments (**h**, **i**) were repeated three times independently with similar results. Source data are provided as a Source Data file.

radiotherapeutic efficacy greatly relies on the damage of biomolecules (particularly DNA) by free radicals and ROS, which were generated from the radiolysis of water, including hydroxyl radical (•OH), hydrogen radical and superoxide anion. The unpaired electrons of these free radicals and ROS induce lesions of biomolecules. For example, pyrimidines of DNA can react with •OH to form a carbon radical in pyrimidines (Fig. 5a). These lesions can be repaired by reductants, such as GSH. Without the reparation, the lesions in biomolecules lead to their structural damage, such as SSBs or DSBs of DNA and cross-linking of DNA to DNA, or DNA to proteins, eventually resulting in cell death. Oxygen is a prototype of radiosensitizers, and its two unpaired electrons can rapidly add to the free radicals in injured biomolecules, consolidating the lesion and boosting the chain reaction of free radicals. Nimorazole is a radiosensitizer as the mimic of oxygen, which contains a nitro group that has electron affinity. The nitro group can also rapidly add to the free radicals of injured biomolecules, consolidating the lesion in a manner similar to oxygen.

We have compared the radiosensitization efficacy and side effects of Mb@Gd-NTs with nimorazole at different doses under RT, and the therapeutic schedule was shown in Fig. 5b. Nimorazole at doses of 125 mg/kg to 250 mg/kg gave obvious and increased radiosensitization effects, as shown by the inhibition of LLC tumor growth (Fig. 5c). However, the increase of nimorazole to a high dose range of 500–750 mg/kg did not further enhance treatment efficacy and no further prolonged survival rate was observed (Fig. 5d) without significant differences ($P = 9505$, Supplementary Table 2). Nimorazole at the highest experimental dose of 750 mg/kg did not completely eliminate the tumors, resulting in tumor recurrence. Although the high dose of 750 mg/kg did not cause obvious mice weight loss (Supplementary Fig. 12a) or hematologic changes (Supplementary Fig. 13), it led to mouse death due to the acute toxicity of nimorazole. On the contrary, Mb@Gd-NTs at dose range of 5 μmol/kg to 20 μmol/kg produced positively correlated RT enhancement (Fig. 5c). Mb@Gd-NTs at the dose of 20 μmol/kg enabled complete tumor ablation without recurrence in 120 days (Fig. 5d). In addition, Mb@Gd-NTs based RT also gave minimal side effects, with no significant changes in body weight (Supplementary Fig. 12b), hematology (Fig. 5e, Supplementary Fig. 14), or histopathology of major organs (Supplementary Fig. 15), indicating the safe profile of Mb@Gd-NT-based RT.

Radiation-induced pneumonitis and enteritis are common side effects of radiotherapy. Mb@Gd-NTs + RT achieved good radiotherapeutic efficacy on LLC tumors, which could possibly be applied to the treatment of different types of cancers such as lung and intestinal cancers. Therefore, the radiation-induced side effects on lungs and intestines were further investigated. C57BL/6 mice were divided into three groups with different treatments including PBS (G1'), PBS + RT (G2'), Mb@Gd-NTs + RT (G3'). Mouse abdomen or chest was respectively irradiated with a dose of 2 Gy for 5 times, and Mb@Gd-NTs were i.v. injected with a dose of 27.15 mg/kg 24 h before each RT session (as shown in the schedule of Supplementary Fig. 16a). Oxidation stress, inflammatory responses, and fibrosis are typical symptoms of radiation-induced pneumonitis and enteritis. To examine the oxidation stress and inflammatory responses of lungs and intestines, the reactive oxygen species (ROS), myeloperoxidase (MPO), and interleukin-1β (IL-1β) were analyzed two weeks after the last radiation. Mb@Gd-NTs + RT had little influence on ROS, MPO, and IL-1β of lungs (Fig. 5f) and intestines (Fig. 5g). Masson staining of tissue section can directly show the percentage and distribution of collagen fibers. Mb@Gd-NTs + RT also had little influence on the collagen fibers of lungs (Fig. 5h, Supplementary Fig. 16b–d), colorectum (Fig. 5i, Supplementary Fig. 17a, b), and small intestines (Supplementary Fig. 17c–f). Therefore, Mb@Gd-NTs + RT could hardly induce pneumonitis, enteritis, and fibrosis.

The radiosensitization efficacy was further evaluated in a clinically relevant human breast cancer orthotopic xenograft model established using a luciferase-transfected MCF-7 cell line (MCF-7 $^{luc+}$). The therapeutic schedule and grouping were shown in Fig. 6a. Tumor growth was monitored by bioluminescence imaging, as shown in Fig. 6b. The radiosensitization effect by Mb@Gd-NTs + RT was significantly higher than the calculated predicted additive effect of Mb@NTs + RT and Gd-NTs + RT, resulting in the lowest tumor bioluminescence signal (Fig. 6c) and the lowest tumor weight at the end of therapy (Fig. 6d). This result further proved that there was an obvious synergic effect between Gd-NTs and $O_2$ ("1 + 1 > 2"). Mb@Gd-NTs + RT achieved better tumor growth inhibition (Fig. 6b–d) and higher overall survival (Fig. 6e) than nimorazole. Mb@Gd-NTs + RT also had minimal influence on mouse body weight (Supplementary Fig. 18), further demonstrating its notable biosafety.

## The long-term immune memory in vivo induced by Mb@Gd-NTs + RT

Mb@Gd-NTs + RT could eradicate LLC tumors (Fig. 5c), leading to the long survival of 100% mice without tumor recurrence (Fig. 5d). Therefore, we examined whether Mb@Gd-NTs + RT could elicit long-term immune memory in vivo to inhibit tumor recurrence. As shown in the scheme (Fig. 7a), for analyzing the long-term immune memory, C57BL/6 mice were inoculated with LLC cells on day -9 to allow a tumor volume of ~ 100 mm³ on day 0, which were randomized into two groups: G2, PBS + RT; G3, Mb@Gd-NTs + RT. Healthy mice without tumor inoculation (naïve mice) were adopted as controls (G1). Tumors of G2 mice were irradiated with a dose of 10 Gy on days 0, 2, 4, 6, and 8 for a total dose of 50 Gy to enable the eradication of tumors. Tumors of G3 mice were irradiated with a dose of 2 Gy on days 0, 2, 4, 6, 8, and 10 a total dose of 12 Gy, and Mb@Gd-NTs were i.v. injected with a dose of 27.15 mg/kg (20 μmol/kg) 24 h before each RT session, which also enabled the eradication of tumors. Immune memory cells were analyzed on day 90. The splenocytes were collected for analyzing their specific cytotoxicity on homogeneous LLC cells and heterogenous MC38 cells. The splenocytes of G3 had specific and significant cytotoxicity against LLC cells, but showed little influence on the viability of MC38 cells (Fig. 7b, c), demonstrating the antigen specificity of the immune response. Noteworthily, the splenocytes of G3 showed significantly higher cytotoxicity on LLC cells compared with the splenocytes of G1 and G2 (Fig. 7b, c), demonstrating Mb@Gd-NTs + RT generated obvious immune memory, which was stronger than PBS + RT. To further prove it, the antigen-specific T cells in splenocytes were quantified by both flow cytometry (Fig. 7d, e, Supplementary Fig. 19) and enzyme-linked immunospot (ELISPOT) assay (Fig. 7f). We found there were more IFN-γ$^+$ cytotoxic T lymphocytes in splenocytes of G3, which was 2.13 and 4.79 times as high as that of G2 and G1 with significant differences (***$P < 0.0001$, Fig. 7e). Furthermore, G3 elicited significantly more central memory T cells ($T_{cm}$) and effector memory T cells ($T_{em}$) both in splenocytes (Fig. 7g, h) and blood (Fig. 7i, j) compared with G1 and G2. For example, the $T_{cm}$ and $T_{em}$ in blood of G2 only increased by 64.6% and 52.9% compared with G1, but the $T_{cm}$ and $T_{em}$ in blood of G3 dramatically increased by 201.9% and 144.1% compared with G1 (Fig. 7j). Therefore, Mb@Gd-NTs + RT could generate obvious antigen-specific CD8$^+$ T cell-mediated immune memory, which was much stronger than PBS + RT.

To further study the long-term immune memory, tumor-free mice of G1, G2, and G3 (obtained on day 90) were re-challenged with LLC or MC38 cells (subcutaneous inoculated with these tumor cells), and tumor growth was recorded for 20 days. G3 protected 100% of mice from the rechallenge of LLC cells, while G2 only protected 33.3% of mice from the rechallenge of LLC cells (Fig. 7k), further demonstrating the more powerful immune memory induced by Mb@Gd-NTs + RT compared with PBS + RT. Meanwhile, both G2 and G3 could hardly inhibit the growth of MC38 cells (Fig. 7l), further proving the antigen-specificity of immune memory.

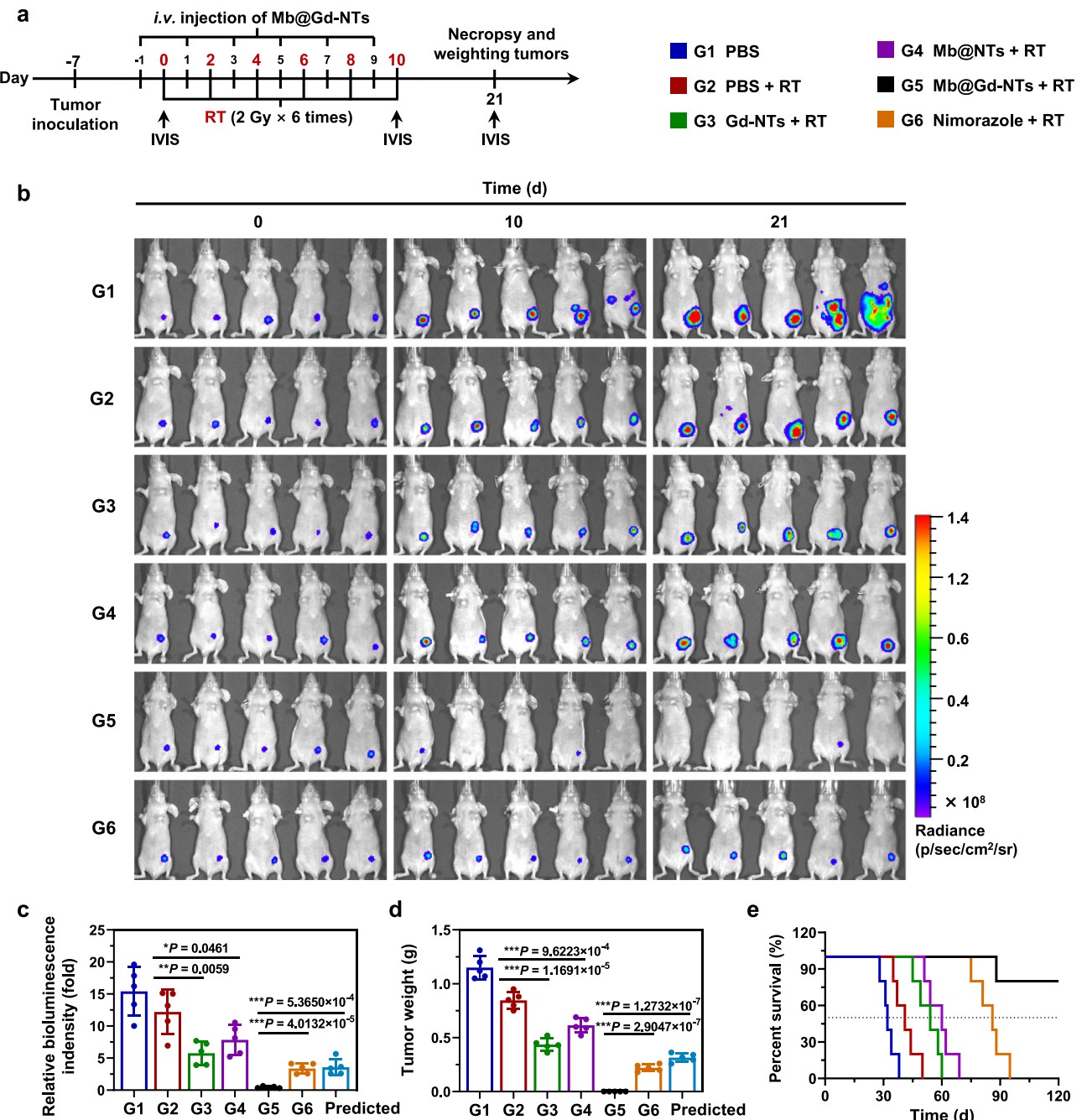

**Fig. 6 | Evaluating the radiosensitization effect of Mb@Gd-NTs on breast orthotopic tumor model. a** Scheme and grouping of in vivo therapy. The breast pad of BALB/c nude mouse was inoculated with $2 \times 10^6$ luciferase-transfected MCF-7 cells (MCF-7 $^{luc+}$) on day -7. The tumor site was irradiated with a dose of 2 Gy on days 0, 2, 4, 6, 8, and 10 for a total dose of 12 Gy. Mb@Gd-NTs were intravenously injected with a dose of 27.15 mg/kg 24 h before each RT session, whereas nimorazole was i.p. injected with different doses of 125–750 mg/kg 30 min before each RT session. Bioluminescence imaging was performed on days 0, 10, and 21 using the IVIS Spectrum in vivo imaging system. **b** Bioluminescence imaging of MCF-7 $^{luc+}$ tumor-bearing nude mice during in vivo therapy (*n* = 5 mice). **c, d** The tumor bioluminescence intensity (**c**) and tumor weight (**d**) measured on day 21. The relative bioluminescence intensity measured on day 21 was normalized to that of day 0. The data (**c, d**) are shown as the mean ± SD (*n* = 5 mice). Statistical analysis was performed by a two-tailed unpaired *t* test. $^*P < 0.05$; $^{**}P < 0.01$; $^{***}P < 0.001$. **e** The survival curves of mice with different treatments (*n* = 5 mice). Source data are provided as a Source Data file.

## Discussion

The small molecular Gd-Tex has limited tumor-targeting ability, and has a quick clearance rate with a short half-life ($t_{1/2\beta} = 16.4$ h) in blood, leading to limited tumor uptake and radiosensitization efficacy, which is one of the main reasons of the failure of its phase III clinical trials[18,19]. However, the nano-formulation Mb@Gd-NTs with PEGylated shell has a decreased clearance rate with a prolonged half-life ($t_{1/2\beta} = 23.98$ h) in blood, which could significantly increase Gd-Tex accumulation in

tumors. Meanwhile, the high-density packing of Gd-Tex in a single Mb@Gd-NTs and the passive tumor-targeting ability of nanovesicles via the EPR effect also largely increased the Gd-Tex accumulation in tumors, contributing to the enhanced radiosensitization efficacy of Mb@Gd-NTs.

The radiosensitization efficacy of Gd-Tex was further increased by investigating its radiosensitization mechanism. In addition to "futile redox cycling", different mechanisms have been proposed to explain

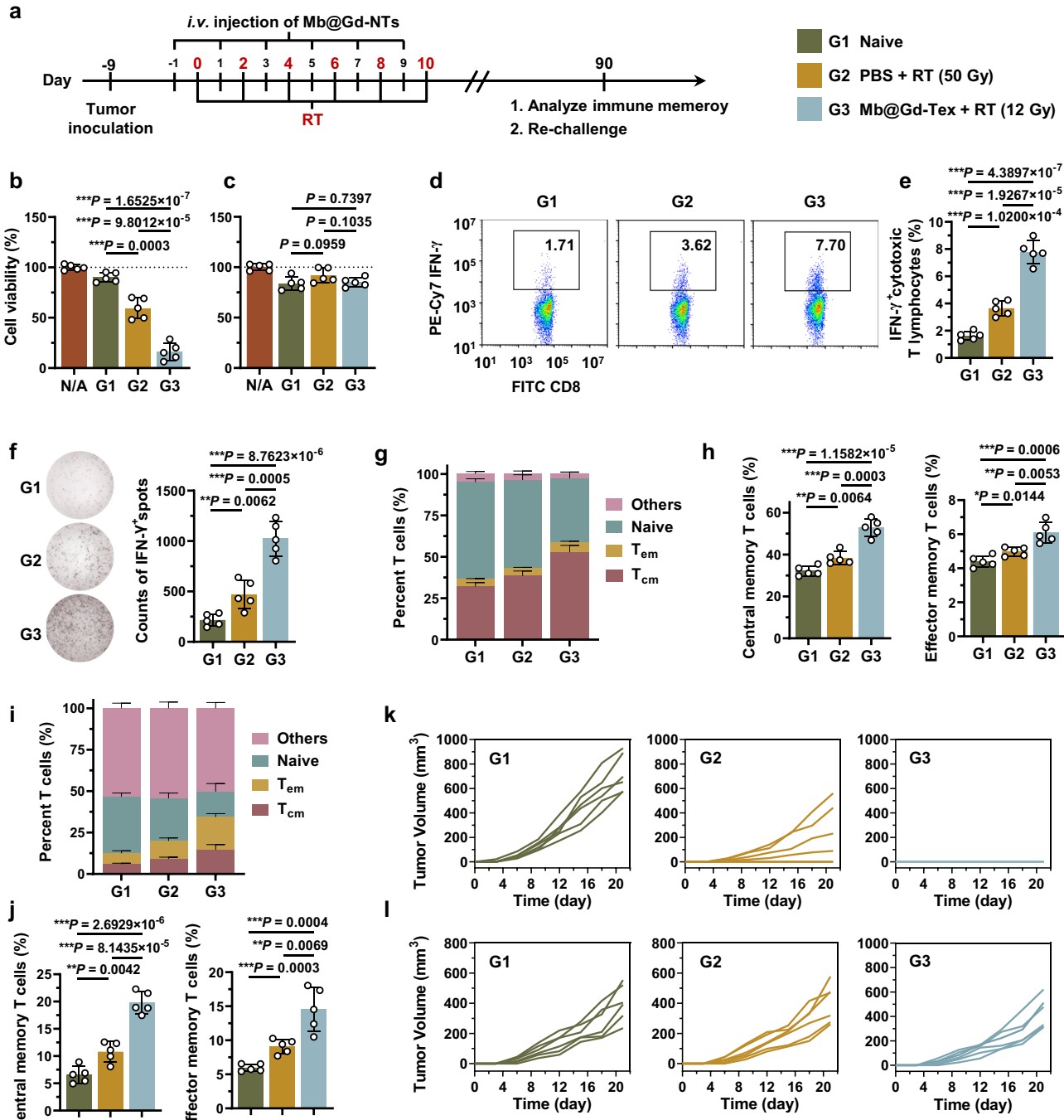

**Fig. 7 | The long-term immune memory in vivo induced by Mb@Gd-NTs + RT.**
**a** Scheme and grouping of the analysis of long-term immune memory. The C57BL/6 mice were inoculated with LLC cells on days −9. For G3 (Mb@Gd-NTs + RT), the tumor sites were irradiated with a dose of 2 Gy on days 0, 2, 4, 6, 8, and 10 for a total dose of 12 Gy, and Mb@Gd-NTs were intravenously injected with a dose of 27.15 mg/kg (20 μmol/kg) 24 h before each RT session. For G2 (PBS + RT), tumors of mice were irradiated with a dose of 10 Gy on days 0, 2, 4, 6, and 8 for a total dose of 50 Gy, enabling the eradication of tumors. Healthy mice without tumor inoculation (naïve mice) were adopted as controls (G1). Immune memory cells were analyzed on days 90. **b**, **c** Cell viability of LLC (**b**) and MC38 cells (**c**) after the incubation with splenocytes collected on days 90 examined by LDH release assay, demonstrating the specific killing ability of splenocytes. **d**, **e** Flow cytometry analysis of IFN-γ⁺

cytotoxic T lymphocytes in splenocytes re-stimulated with whole cell lysate of LLC cells. **f** IFN-γ secretion by the splenocytes, as determined by ELISPOT assay after re-stimulation with whole cell lysate of LLC cells. **g**–**j** Quantitative analysis of effector memory T cells ($T_{em}$, CD3⁺CD8⁺CD62L⁻CD44⁺), central memory T cells ($T_{cm}$, CD3⁺CD8⁺CD62L⁺CD44⁺) and *naive* T cells ($T_{naive}$, CD3⁺CD8⁺CD62L⁺CD44⁻) in splenocytes (**g**, **h**) and blood (**i**, **j**), showing the radiosensitization-elicited T cell memory. **k**, **l** Tumor growth curves of LLC (**k**) and MC38 (**l**). Tumor-free mice (*n* = 6 mice) of G1, G2, and G3 (obtained on days 90) were re-challenged with LLC or MC38 cells, and the tumor growth was recorded for 20 days. The data (**b**, **c**, and **e**–**j**) are shown as the mean ± SD (*n* = 5 mice). Statistical analysis was performed by a two-tailed unpaired *t* test (**b**, **c**, **e**, **f**, **h**, **j**). *P < 0.05; **P < 0.01; ***P < 0.001. Source data are provided as a Source Data file.

the radiosensitization effect of Gd-Tex, including the oxidization of thioredoxin reductase, the interference of ribonucleotide reductase and DNA repair, the mobilization and release of intracellular zinc, as well as the promotion of the mitochondrial apoptosis pathway[18]. These different mechanisms also rely on the oxidation capability and ROS generation of Gd-Tex. Therefore, "futile redox cycling" is the most important and basic mechanism. The hypothesis of "futile redox cycling" can be divided into two steps, as shown in Fig. 4h[22]. In the first step, Gd-Tex can oxidize cellular reducing metabolites (such as ascorbate, NADPH, and reduced glutathione) and generate Gd-Tex radicals (Gd-Tex·). Since these reduced metabolites can help tumor cells repair the damage caused by RT, Gd-Tex attenuates the resistance of tumor cells to RT. In the second step, Gd-Tex· reacts with $O_2$ and generates superoxide anions, which are transformed into hydrogen peroxide under the catalysis of superoxide dismutase. Superoxide anions and hydrogen peroxide belong to the ROS family. Thus, Gd-Tex increased the intracellular ROS level of tumor cells and further weakened the radiation resistance of tumor cells. However, "futile redox cycling" is only a hypothesis, and although $O_2$ is assumed to be involved in this hypothesis, little experiment has been done to confirm the $O_2$ dependence of Gd-Tex radiosensitization.

To elucidate the $O_2$ dependence of Gd-Tex radiosensitization, $O_2$-carriable Mb was loaded into Gd-NTs. The radiosensitization effects of Gd-NTs and MbO$_2$@Gd-NTs were studied and compared in detail. Compared with Gd-NTs, MbO$_2$@Gd-NTs had a higher capability of oxidizing ascorbate and increasing cellular ROS level, inducing a higher degree of DSB and apoptosis in vitro, eventually leading to higher deprivation of viability and colony formation capability of tumor cells. By statistical analysis of in vivo antitumor efficacy, it was found that the radiosensitization effect of Mb@Gd-NTs was superior to the calculated predicted additive effect of Mb@NTs and Gd-NTs ("1 + 1 > 2"), thus confirming the synergistic effect between Gd-Tex and $O_2$. These in vitro and in vivo studies showed Gd-NTs have a radiosensitization effect through a "futile redox cycling", and this effect is indeed dependent on $O_2$ concentration. Relieving tumor hypoxia can effectively improve the radiosensitization efficacy of Gd-NTs.

Besides influencing the radiosensitization efficacy of Gd-NTs, tumor hypoxia can also directly influence the RT efficacy. Tumor hypoxia is one of the basic characteristics of the tumor microenvironment[30]. As a hostile hallmark, tumor hypoxia is related to poor prognosis, tumor metastasis and resistance to various treatments[23,31]. Hypoxia is particularly closely related to the RT efficacy and is the major reason for the ineffectiveness of RT, as oxygen concentration directly influences the generation of cytotoxic free radicals of radiation[32,33]. Oxygen carrier protein Hb has been proposed for improving hypoxia and RT efficacy, but its efficacy is limited by rapid degradation in blood[13,34,35]. PEGylated Hb conjugated with polyethylene glycol has undergone human clinical trials of radiosensitization[36]. Herein, PEGylated stealth nanoplatform Gd-NTs efficiently prolonged the in vivo half-life of Hb and Mb, increased tumor accumulation, and finally improved the efficacy of relieving tumor hypoxia and the radiosensitization, which increased the potential of the clinical application of Mb/Hb as radiosensitizers. It was also found that Mb had a more powerful capability than Hb to relieve tumor hypoxia due to its special oxygen affinity.

The capability of Mb@Gd-NTs nanoplatform to chelate various trivalent metal ions enabled dual-modality imaging-guided therapy. Nuclear medicine imaging of SPECT is highly sensitive and quantitative, which can accurately show the in vivo behavior of radiolabeled compounds, even if the concentration of the radiolabeled compounds in tissue is very low[37,38]. MRI can cooperate with SPECT imaging to provide detailed anatomical information[39]. Positron emission tomography (PET)/MRI based on an $^{18}$F-FDG imaging probe has been applied in clinical practice[40]. SPECT/MRI dual-modality imaging based on Mb@$^{177}$Lu/Gd-NTs can not only clearly show the anatomical location of tumors, but can also quantitatively display the drug biodistribution in real time, which is helpful to determine the administration time of external beam irradiation.

In summary, the nanoplatform Mb@Gd-NTs, based on the self-assembly of Gd-Tex-lipid was constructed to improve the radiosensitization effect of Gd-Tex. The high-density packing of Gd-Tex in a single nanovesicle, the passive tumor-targeting ability of nanovesicles and the prolonged half-life in blood significantly increased Gd-Tex accumulation in tumors. The $O_2$ dependence of the radiosensitization effect of Gd-Tex was confirmed, and the nanoplatform enabled spatiotemporal codelivery of $O_2$ and Gd-Tex, achieving the relief of tumor hypoxia and significant enhancement of Gd-Tex radiosensitization, eventually inducing the obvious long-term antitumor immune memory to inhibit tumor recurrence. The loading of Mb into the nanoplatform prolonged its half-life and improved its efficacy in relieving hypoxia. The simply constructed and highly biocompatible "one-for-all" nanoplatform with radiosensitization and SPECT/MRI imaging functions exhibits promising clinical application for imaging-guided radiotherapy.

## Methods

### Study design

Our research complies with all relevant ethical regulations. The study protocol was approved by local ethics committees. All animal studies were performed in accordance with ARRIVE guidelines. All animal experiments were approved by the Institutional Animal Care and Use Committee at the Institute of Biophysics, Chinese Academy of Science. The animal study complied with relevant ethical regulations for animal testing and research. The objective of this study was to investigate the $O_2$-dependence of the radiosensitization effect of Gd-Tex, and to determine whether the nanovesicle Mb@Gd-NTs could improve the radiosensitization effect of Gd-Tex by increasing Gd-Tex accumulation in tumor and enabling spatiotemporal codelivery of $O_2$ and Gd-Tex into tumors. Both in vitro and in vivo assays were performed to elucidate the impact of $O_2$ concentration on Gd-Tex radiosensitization. For in vitro assays, 3-5 replicates within each condition were used to ensure statistical power, which could sufficiently represent intragroup variations and allow for defining statistical significance. For in vivo antitumor experiments of subcutaneous xenograft animal model, mice were randomized on the basis of the tumor size before administration to ensure similar average tumor sizes across groups, and five mice were included within each group to ensure statistical power, which enabled us to statistically distinguish tumor sizes and survival rates across groups. For in vivo antitumor experiments of the orthotopic xenograft model, tumor formation was monitored by bioluminescence using the IVIS Spectrum in vivo imaging system. Mice were randomized on the basis of the bioluminescent imaging before administration to ensure similar average tumor sizes across groups. SPECT, MRI and fluorescence imaging were employed to track the in vivo behavior of nanovesicles, and the signal intensity of the region of interest (ROI) was quantified using corresponding software of SPECT, MRI and fluorescence imaging, and at least three mice were included within each group to ensure statistical power. The investigators performing the animal experiments were not blinded to group information.

### Materials

Gadolinium acetate hydrate and skeletal muscle Mb were purchased from Sigma–Aldrich (USA). Triethylamine was purchased from Alfa Aesar (USA). Distearoylphosphatidylcholine (DSPC) and DSPC-polyethylene glycol 2000 (DSPE-mPEG2000) were purchased from Avanti Polar Lipids (USA). Cholesterol was purchased from Tokyo Chemical Industry (Japan). Cell culture dishes and plates were purchased from Corning (USA). Mouse IFN-γ ELISA Kit (Catalog No. CZM10-96) was purchased from Beijing CHENG ZHI KE WEI Biotechnology Co., Ltd. (China). Mouse IL-1β ELISA Kit (Catalog No.

EMC001bQT) was purchased from NeoBioscience Technology Co., Ltd. (China). Ultrapure water was obtained using a Millipore Milli-Q Gradient System (USA). Rabbit anti-γ-H2A.X (phospho S139) antibody (Catalog No. ab81299, Clone: EP854(2)Y, 1: 250) was purchased from Abcam (USA). FITC-conjugated goat anti-rabbit IgG H&L antibody (Catalog No. ab6717, 1: 1000) was purchased from Abcam (USA). APC anti-mouse CD3 antibody (Catalog No. 100312, Clone:145-2C11, 1: 100), FITC anti-mouse CD8 antibody (Catalog No. 100706, Clone: 53-6.7, 1: 100), PE/Cyanine7 anti-mouse IFN-γ antibody (Catalog No. 505826, Clone: XMG1.2, 1: 100), PE anti-mouse/human CD44 antibody (Catalog No. 103008, Clone: IM7, 1: 100), and PE/Cyanine7 anti-mouse CD62L antibody (Catalog No. 104418, Clone: MEL-14, 1: 100) were purchased from BioLegend (USA).

### Synthesis and purification of Gd-Tex-lipid

Tex-phospholipid conjugate (Tex-lipid) was synthesized according to the literature[20,21]. In a 5 mL round bottom flask, 10 mg of Tex-lipid, 6 mg of gadolinium acetate hydrate, and 9 mg of triethylamine were dissolved into 1 mL of methanol. The solution was heated and stirred at 50 °C for 3 h. The solvent was removed under a high vacuum, and the residue was washed with 10 mL hexanes three times, yielding a dark green crude product. The crude product was purified using a flash chromatography silica gel column. An initial gradient of methanol: dichloromethane (0-25% over 10 min) was used, followed by a second gradient of chloroform:methanol:water (35:14:1, isocratic over 15 min). This second gradient was used for moieties such as phosphocholine or functional groups that are stuck to silica gel. The purified Gd-Tex-lipid was verified by MALDI-TOF-MS and ESI-MS.

### Fabrication of Gd-NTs and Mb/Hb@Gd-NTs

A typical filming-rehydration method was adopted. 2 mg of Gd-Tex-lipid, 1.032 mg of DSPC, 0.424 mg of cholesterol, and 0.614 mg of DSPC-polyethylene glycol 2000 (DSPE-mPEG2000, purchased from Xi'an ruixi Biological Technology Co., Ltd, China) were dissolved into 1 mL of chloroform, and the molar ratio of each component was 40:30:25:5. The solvent was dried under the vacuum in a rotary evaporator (IKA, Germany) to form a film, and the film was further dried for 12 h to remove traces of chloroform. The film was hydrated with PBS solutions of Mb or Hb, heated at 40 °C for 30 min, and then sonicated in a water bath with 100% output power for 30 min to obtain multilamellar liposomes (MLVs). Uniform unilamelar liposomes (ULVs) were obtained using a high-pressure extruder (Northern Lipid Inc., USA), and the MLV solution was extruded 3 times with a 100 nm double-layer polycarbonate membrane (Whatman, USA) at 40 °C. The unencapsulated Mb/Hb was removed using a dialysis bag with a molecular cutoff of 1000 kDa and dialyzing ULV solution against PBS solutions for 72 h. The protein concentrations of Mb and Hb were measured using BCA (bicinchoninic acid) Protein Assay Kit (Catalog No. PA002; Novoprotein, China).

### Synthesis of $^{177}$Lu-Tex-lipid and fabrication of Mb@$^{177}$Lu/Gd-NTs

For the synthesis of $^{177}$Lu-Tex-lipid, 0.5 mg of Tex-lipid, 100 µCi $^{177}$LuCl$_3$, and 0.45 mg of triethylamine were dissolved into 100 µL of methanol in a 1.5 mL Eppendorf tube. The solution was heated at 50 °C for 20 min, yielding $^{177}$Lu-Tex-lipid. For the fabrication of Mb@$^{177}$Lu/Gd-NTs, 1.5 mg of Gd-Tex-lipid, 0.774 mg of DSPC, 0.318 mg of cholesterol, and 0.461 mg of DSPE-mPEG2000 were dissolved into 0.5 mL of chloroform, followed by the addition of the synthesized $^{177}$Lu-Tex-lipid. The solvent was dried by heating, and the formed lipid film was hydrated with 0.5 mL of PBS solutions of Mb at 40 °C for 30 min and then sonicated in a water bath with 100% output power for 30 min, followed by filtration through a 0.22 µm syringe filter (Millipore, USA) to remove the large MLVs. The free $^{177}$LuCl$_3$ and unencapsulated Mb were removed using Amicon® Ultra-0.5 centrifugal filter devices (Millipore, USA) with a nominal molecular weight limit (NMWL) of 100 kDa.

### Characterization of nanoparticles

The morphology of liposomes was identified by transmission electron microscopy. The carbon-coated copper mesh was treated with glow discharge for 5 min, and the nanoparticles were diluted to 0.05 mg/mL, dropped on the copper mesh and incubated for 2 min. The copper mesh was washed three times with ultrapure water, stained with 2% uranyl acetate, and observed on a Tecnai F20 electron microscope (FEI, USA) with a voltage of 200 kV. The image was photographed by F114 CCD (TVIPS). The size distribution of nanoparticles diluted in PBS was characterized by a dynamic light scattering instrument (Wyatt, USA). The UV–Vis absorption spectra were measured using a multi-functional microplate reader (BioTek, USA).

### Evaluation of the oxygen-carrying capacity of nanoparticles

The oxygen release curve of MbO$_2$@Gd-NTs was monitored using a portable dissolved oxygen meter (Rex, JPBJ-608, China). Briefly, Mb@Gd-NTs were saturated with oxygen flow (5 L/min) for 10 min, yielding MbO$_2$@Gd-NTs. The detection probe of the dissolved oxygen meter was immersed in 3 mL of PBS buffer, followed by the quick injection of 500 µL of MbO$_2$@Gd-NTs with Mb concentrations of 0, 10, 20, 40, and 80 µg/mL. The concentration of dissolved oxygen was measured every second for 10 min. The oxygen release curves of free HbO$_2$, free MbO$_2$, HbO$_2$@NTs, and MbO$_2$@NTs were monitored and compared using the same method.

### Evaluation of the capability of Gd-NTs to oxidize ascorbate

A total of 100 µL of NTs, MbO$_2$@NTs, Gd-NTs, or MbO$_2$@Gd-NTs in an aqueous solution was added to a quartz cuvette containing sodium ascorbate solution (50 mM HEPES, 100 mM NaCl, pH = 7.5). The absorbance of the reaction solution at 266 nm (the characteristic absorption peak of sodium ascorbate) was monitored using a UV–Vis absorption spectrophotometer (Hitachi, Japan) during the reaction. From the decrease in absorbance, the cumulative oxidation percentage of sodium ascorbate was calculated according to the literature[22].

### Cell culture and animals

The LLC (Catalog No. CRL-1642) and MCF-7 (Catalog No. HTB-22) cell lines were originally obtained from American Type Culture Collection (ATCC; Manassas, VA, USA). The MC38 cell line (resource No. 1101MOU-PUMC000523) was obtained from the Cell Resource Center, Peking Union Medical College (which is the headquarter of National Science & Technology Infrastructure, National BioMedical Cell-Line Resource, NSTI-BMCR, China). All cell lines are not listed in the database of commonly misidentified lines maintained by the International Cell Line Authentication Committee. All cell lines were carried out with mycoplasma detection and were negative for mycoplasma contamination using Quick Cell Mycoplasma Assay kit (Catalog No. AC16L061, Life-iLab, China). LLC cells were cultured in Dulbecco's modified Eagle medium (DMEM) with 4.5 g/L glucose (Corning, USA). MC38 cells were cultured in RPMI-1640 medium. MCF-7 cells were cultured in MEM-EBSS (MEM Eagles with Earle's Balanced Salts). The cells were cultured with 10% fetal bovine serum (Gibco, USA), 100 U mL$^{-1}$ penicillin and 100 µg mL$^{-1}$ streptomycin (Catalog No. 60162ES76; Yeasen, Shanghai, China) at 37 °C. Normoxic cells were maintained in a humidified atmosphere containing 20% O$_2$ and 5 vol% CO$_2$, and hypoxic cells were maintained in a humidified atmosphere containing 1% O$_2$ and 5 vol% CO$_2$. Luciferase-expressing MCF-7 (MCF-7 $^{luc+}$) cells were obtained by stably transfecting MCF-7 cells with the encoding gene of luciferase (Gene ID: 116160065) using FectinMore™ Transfection Reagent (Catalog No. CM001, Chamot Biotechnology Co., Ltd., China).

All animal experiments were approved by the Institutional Animal Care and Use Committee at the Institute of Biophysics, Chinese Academy of Science. The LLC xenograft tumor model was established by inoculating $4 \times 10^6$ LLC cells into the right hind thigh of 5-week-old

female C57BL/6 mice obtained from Vital River Laboratory Animal Technology Co. Ltd (China). Mice were housed in individually ventilated cages (IVC) with sterile ventilation systems at a temperature of 20-22°C and a humidity of 30–70%. Feed and water were available ad libitum. Artificial light was provided in a 12 h light/12 h dark cycle. The maximal tumor size permitted by the ethics committee is 2000 mm³, and the maximal tumor size was not exceeded in this study.

## Detection of cell hypoxia in vitro

LLC cells were seeded in a 35 mm confocal dish with a 15 mm bottom well (Cellvis, USA) at a density of $10^5$ cells per dish and cultured at 20% $O_2$ until cell adherence was ensured. Cells were further cultured in 1% $O_2$ for 24 h to induce cells to become hypoxic. The culture medium was replaced with various sterile agents dissolved in 1 mL of DMEM, including NTs, Gd-NTs, $MbO_2$, $MbO_2$@NTs, $MbO_2$@Gd-NTs, $HbO_2$, and $HbO_2$@NTs. $MbO_2$/$HbO_2$-containing agents were prepared from Mb/Hb-containing agents saturated with oxygen. After incubation at 1% $O_2$ for 4 h, the cells were repeatedly incubated with the same agent for 4 h once again. Cell hypoxia status was detected using Hypoxyprobe™-1 plus Kit (Hypoxyprobe Inc., USA) by immunofluorescence staining according to the manufacturer's instructions, and the nucleus was stained with DAPI, followed by observation under a confocal microscope (Zeiss LSM 700, Germany).

## Detection of cellular ROS, apoptosis, DSB, viability and colony formation capability

LLC cells were seeded in a 24-well plate at a density of $10^5$ cells, cultured at 20% $O_2$ for cell adherence, and further incubated at 1% $O_2$ for 24 h to induce cells to become hypoxic. Cells were incubated with various nanoparticles dissolved in 1 mL of DMEM for 4 h at 1% $O_2$ and repeatedly incubated with the same nanoparticles for 4 h once again. Cells were irradiated with an X-ray irradiator (Rad Source RS2000XE, USA) at a dose of 2 Gy, followed by different assays(1). Cellular ROS levels were detected using the H2DCFDA probe (Reactive Oxygen Species Assay Kit, Catalog No.O040, LABLEAD Inc., China). After X-ray irradiation, the cells were incubated with 10 μM H2DCFDA for 30 min, digested with 0.05% trypsin (ZOMANBIO, China), and collected to analyze the fluorescence intensity of H2DCFDA using a flow cytometer (BD FACSCalibur, USA). Cellular $H_2O_2$ levels were detected using $H_2O_2$ assay kit (Geruisi-bio, China)(2). Cell apoptosis was examined using an Annexin V-PI double staining kit (Solarbio, China). The adherent and floating cells were collected 12 h after X-ray irradiation, stained with Annexin V and PI according to the manufacturer's instructions, and analyzed using a flow cytometer(3). Cellular DSB was detected by immunofluorescence staining of γ-H2A.X. Cells were fixed with 4% paraformaldehyde 30 min after X-ray irradiation, permeated with 0.1% Triton-100, blocked with 3% BSA solution (dissolved in 0.1% PBST) for 1 h, incubated with rabbit anti-human γ-H2A.X primary antibody (Abcam, USA) or Rabbit Control IgG (Catalog No. CR1; Sino Biological Inc., China) at 4 °C overnight, washed with 0.1% PBST, incubated with FITC-conjugated anti-rabbit secondary antibody for 1 h, stained with DAPI, and observed under a confocal microscope(4). Cell viability was detected by CCK-8 assay kit (KeyGEN, China). In contrast, cells were seeded in a 96-well plate at a density of $10^4$ cells and treated as above. Twenty-four hours after X-ray irradiation, 10 μL of CCK-8 was added to each well for incubation for 30 min. The absorbance of each well at 450 nm was measured using a microplate reader(5). Differently, cells were irradiated with X-rays at doses of 2, 4, 6, and 8 Gy. After irradiation, the cells were collected, and 1% of the cells were seeded in a Petri dish with a 6 cm diameter and cultured in 20% $O_2$ for 10 days. Cells were fixed with 4% paraformaldehyde and stained with 1% crystal violet for 5 min. The number of cell colonies in each dish was counted to calculate the survival fraction.

"Predicted additive effect" in Fig. 2i means the calculated theoretical additive radiosensitization effect of Gd-NTs and $MbO_2$@NTs

(assuming there was no synergistic effect), which was calculated by multiplying the survival inhibition ratios of Gd-NTs + RT and $MbO_2$@NTs + RT according to the reported method[41].

## Fluorescence imaging and pharmacokinetics studies of Cy5.5-labeled Mb@Gd-NTs

A pharmacokinetics study was performed on 5-week-old C57BL/6 mice. Mice were divided into 6 groups (n = 5) and intravenously injected with 200 μL of free Mb, free Hb, Mb@NTs, Mb@Gd-NTs, Hb@NTs, or Hb@Gd-NTs. Both Mb and Hb were labeled with Cy5.5 fluorescent dye. Fifty microliters of blood were sampled from the orbital vein before and 15 min, 30 min, 1 h, 2 h, 4 h, 8 h, 18 h, 30 h, and 48 h after injection. The blood was added to 100 μL of EDTA-2Na aqueous solution and stored at 4 °C. After sampling, 100 μL of each blood sample was added to an opaque black 96-well plate. The fluorescence intensity of Cy5.5 was detected by a multifunctional microplate reader, and the blood concentration (ID%/mL) was calculated from the standard curve. The standard curve was obtained by dissolving a gradient concentration of Cy5.5-labeled Mb/Hb in mouse blood ($\lambda_{ex}$ = 678 nm, $\lambda_{em}$ = 694 nm). Pharmacokinetic parameters were calculated using PKslover software and a bicompartmental model[42].

Fluorescence imaging and biodistribution studies were also performed on C57BL/6 mice. At 0.5, 2, 4, 8, 12, 24, 48, and 72 h after injection, in vivo fluorescence imaging was performed using the IVIS Spectrum in vivo imaging system (PerkinElmer, USA). Two hours after the injection of Mb and 24 h after the injection of Mb@NTs, Hb@NTs, and Mb@Gd-NTs, organs (lung and muscle) and tumors were collected, weighed, homogenized by a tissue homogenizer, and lysed with RIPA Lysis Buffer (Catalog No. abs9229; Absin, China) plus ultrasonication. The fluorescence intensity of Cy5.5 was measured by a multifunctional microplate reader. The content of Mb/Hb (ID%/g) was calculated from the corresponding standard curves.

## SPECT/CT imaging and biodistribution studies of Mb@¹⁷⁷Lu/Gd-NTs

Small animal SPECT/CT imaging of LLC tumor-bearing mice was performed using a nanoScan® SPECT/CT system (Mediso Ltd, Hungary). Each mouse was intravenously injected with 400 μCi Mb@¹⁷⁷Lu/Gd-NTs. At 2, 12, 24, 48, and 72 h post-injection, the mice were anesthetized by inhaling 1.5% isoflurane, followed by the acquisition of SPECT/CT images. In a 256 × 256 acquisition matrix, a total of 24 projections were acquired for SPECT imaging with a minimum of 50,000 counts per projection. For the reconstruction of SPECT images, an ordered-subset expectation maximization (OSEM) algorithm was used. Prior to each SPECT acquisition, cone beam CT images were acquired using the nanoScan® SPECT/CT system (180 projections, 1 s/projection, 45 kVp). The fusion images of SPECT and CT were obtained using the automatic fusion feature of Nucline software V.2.0 (Mediso Ltd., Hungary).

Biodistribution studies were performed on LLC tumor-bearing mice with a tumor volume of ~ 100 mm³. Mice were intravenously injected with 50 μCi of Mb@¹⁷⁷Lu/Gd-NTs and sacrificed at 6, 24, 48, and 72 h postinjection. Blood, heart, liver, spleen, lung, kidney, stomach, small intestine, large intestine, bone, knee joint, muscle, and tumor samples were harvested and weighed. The radioactivities of the samples were counted using an automatic γ-counter (PerkinElmer, USA). The tissue uptake of Mb@¹⁷⁷Lu/Gd-NTs was calculated as a percentage of the injected dose per gram of tissue (%ID/g).

## Evaluation of MRI

The MRI system (Bruker, Germany) had a spatial resolution of 1 × 1 × 1.5 mm and a field of view of 110×80 mm for coronal scans. The $T_1$-weighted imaging used an effective echo time of 9.5 ms and a repetition time of 1000 ms. The relaxation rate ($R_1$) of different concentrations of Mb@Gd-NTs (0 - 1 mM) was measured at a magnetic field strength of 7 T. Mb@Gd-NTs were dissociated with 0.5% Triton X-100,

and $R_1$ was also measured. MRI in vivo was performed on LLC-bearing mice with a tumor size of ~ 100 mm$^3$ before and 4, 24, 48 h after the injection of Mb@Gd-NTs.

## In vivo evaluation of radiosensitization effect

LLC-bearing C57BL/6 mice with a tumor volume of ~100 mm$^3$ were divided into 9 groups (n = 5) with different treatments: (1) PBS, (2) NTs, (3) Gd-NTs, (4) PBS + RT, (5) Free Mb + RT, (6) NTs + RT, (7) Mb@NTs + RT, (8) Gd-NTs + RT, and (9) Mb@Gd-NTs + RT. Free Mb was intravenously injected 2 h before RT ("RT"), whereas other agents were intravenously injected 24 h before RT. The dose was 5 μmol/kg Tex (6.79 mg/kg) with a volume of 200 μL, and the tumor site was irradiated with a dose of 2 Gy. The first RT was defined as day 0, and RT was conducted on days 0, 2, 4, 6, and 8. The agent was administered before each RT session. Tumor volumes were measured and calculated as $V = (L \times W^2)/2$, where $L$ and $W$ are the length and width of the tumor, respectively.

At the end of therapy (day 16), the mice were sacrificed, and the tumor tissues were collected, fixed in 10% formalin, embedded in paraffin, sectioned and stained with H&E (hematoxylin&eosin). The pathological changes in the tumor tissues were analyzed using an optical microscope. The tumor tissues were also embedded in OCT (Leica, Germany), frozen with liquid nitrogen, and cryosectioned into 5 μm sections. DNA SSBs were detected using a One-step TUNEL In Situ Apoptosis Kit (Elabscience, China) and observed using a confocal microscope. DNA DSBs were examined by immunofluorescence staining of γ-H2A.X using similar procedures mentioned above.

Body weight was monitored every two days. On days 0, 5, 11, 18, 20 μL of blood was collected from each mouse through the orbit for routine blood examination. At the end of therapy, the mice were sacrificed, and the heart, liver, spleen, lung, and kidney were collected, fixed in 10% formalin, sectioned and stained with H&E. The pathological changes in the main organs were observed.

The radiosensitization effect of Mb@Gd-NTs and nimorazole was compared in LLC-bearing mice with a tumor volume of ~ 100 mm$^3$. Different concentrations of nimorazole (125-750 mg/kg) were intraperitoneally injected 30 min before each RT, and different concentrations of Mb@Gd-NTs (5-20 μmol/kg Tex-Lipid) were intravenously injected 24 h before each RT. Six RT sessions were administered at a dose of 2 Gy for a total dose of 12 Gy.

An orthotopic xenograft breast tumor model was established in BALB/c nude mice at 5 weeks old. A total of $2 \times 10^6$ luciferase-transfected MCF-7 cells were inoculated into the mouse breast pad of the left lower abdomen. From the 7th day after inoculation, the tumor site was irradiated with an X-ray dose of 2 Gy every two days for a total of 5 times. Various agents were intravenously injected with a Tex-Lipid dose of 20 μmol/kg 24 h before each irradiation. Mice were intraperitoneally injected with D-luciferin potassium salt at a dose of 150 mg/kg 10 min before each bioluminescence imaging using the IVIS Spectrum in vivo imaging system.

"Predicted additive effect" of Fig. 4a–c and Fig. 6c, d means the calculated additive radiosensitization effect of Gd-NTs and Mb@NTs, assuming there was no synergistic effect between Gd-NTs and Mb@NTs, which was calculated by multiplying the tumor growth inhibition ratios of Gd-NTs + RT and Mb@NTs + RT, according to the reported method[41].

## Analysis of immune memory

The C57BL/6 mice were inoculated with LLC cells ($2 \times 10^6$ cells/mouse) on day -9. For Mb@Gd-NTs + RT, the tumor sites were irradiated with a dose of 2 Gy on days 0, 2, 4, 6, 8, and 10 for a total dose of 12 Gy, and Mb@Gd-NTs were intravenously injected with a dose of 27.15 mg/kg (20 μmol/kg) 24 h before each RT session. For PBS + RT, tumors of mice were irradiated with a dose of 10 Gy on days 0, 2, 4, 6, and 8 for a total dose of 50 Gy, enabling the eradication of tumors. Healthy mice without tumor inoculation (naïve mice) were adopted as controls. Peripheral blood and splenocytes were obtained from mice on day 90. The antigen-specific CD8$^+$ T cells (IFNγ$^+$ cytotoxic T lymphocyte cells) in splenocytes were analyzed by flow cytometry and a mouse IFN-γ precoated ELISPOT kit (Dakewe Biotech Co., Ltd., China) according to the manufacturer's instructions. The memory T cells in peripheral blood and splenocytes were analyzed by flow cytometry, which were categorized as $T_{naive}$ (CD3$^+$CD8$^+$CD44$^-$CD62L$^+$), $T_{cm}$ (CD3$^+$CD8$^+$CD44$^+$CD62L$^+$), and $T_{em}$ (CD3$^+$CD8$^+$CD44$^+$CD62L$^-$). To analyze the specific killing ability of splenocytes, splenocytes were cultured with LLC or MC38 cells at the ratio of 10:1 for 48 h, and the cell viability of LLC and MC38 cells was examined by LDH Cytotoxicity Assay Kit (Yeasen, China). Mice were re-challenged with LLC or MC38 cells ($2 \times 10^6$ cells/mouse, subcutaneous inoculation) on days 90. Tumor volumes were measured and calculated as $V = (L \times W^2)/2$, where $L$ and $W$ are the length and width of the tumor, respectively.

## Statistical analysis

The results are expressed as the mean ± standard deviation (SD). The sample size for analysis was as annotated in figure legends. The differences between two groups were assessed by two-tailed and unpaired $t$ tests. Two-sided log-rank tests were used to analyze Kaplan-Meier curves. $P^* < 0.05$, $P^{**} < 0.01$ and $P^{***} < 0.001$ were considered significant. NovoExpress (version number, 1.4.1.1901) and BD FACS-Calibur Software (version number, 1.0.264.21) were used to collect data of flow cytometry. Living Image software (version number, 4.3.1.16427) was used for the bioluminescence and fluorescence imaging. ZEN 2012 (version number, 1.1.13346.204) was used for the immunofluorescence detection. ImageJ (version number, 1.8.0) was used for the semi-quantification of immunofluorescence images. FlowJo (version number, 10.0.0.0), NovoExpress (version number, 1.4.1.1901) and BD FACSCalibur Software (version number, 1.0.264.21) were used to analyze the data of flow cytometry. GraphPad Prism (version number, 8.3.0.538) and IBM SPSS Statistics (version number, 19.0) were used for the statistical analysis.

## Reporting summary

Further information on research design is available in the Nature Portfolio Reporting Summary linked to this article.

# Data availability

The authors declare that all the data supporting the results in this study are available within the Article, Supplementary Information or Source Data file. The gating strategy for flow cytometry experiments can be found in the Supplementary Information. Source data are provided with this paper.

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

## Acknowledgements

This research was supported by grants from the National Natural Science Foundation of China (NSFC, Grant Nos. 92159201 and 81927802 to F.W., 81971676 to J.S.), the Emergency Key Program of Guangzhou Laboratory (Grant No. EKPG21-16 to F.W.), the Youth Innovation Promotion Association of Chinese Academy of Sciences (YIPACAS, Grant No. 2016090 to J.S.), the Canada Research Chairs Program, Princess Margaret Cancer Foundation, and Terry Fox Research Institute PPG#1075 to G.Z.

## Author contributions

F.W., J.S., and G.Z. conceptualized the project, designed the experiments, acquired the fundings, edited the manuscript and jointly supervised this work. X.M., X. Liang, M.Y., Y.G., Q.L., and X. Li. conducted the experiments and wrote the manuscript. X.M., Y.Y., Y.S., M.C., and J.C. analyzed the data and edited the manuscript.

## Competing interests

The authors declare no competing interests.
