## [Peer Review File · Nature Communications]

REVIEWER COMMENTS

Reviewer #1 (Remarks to the Author):

This study reported a lipid nanovesicles (Mb@Gd-NTs) assembled by the gadolinium-coordinated texaphyrin-lipid (Gd-Tex-lipid) that encapsulate oxygen-binding myosin for radiosensitization. The nanovesicles possessed high loading efficiency of Gd-Tex and long half-life in blood circulation, contributing to the significantly increased Gd-Tex accumulation in tumor compared with free Gd-Tex. The oxygen association of myosin endowed dynamic release of oxygen, improving the tumor oxygenation for the benefit of radiotherapy. Tex was also chelated with radioisotopes (lutetium, ^{177}Lu) that endowed Gd-NTs with SPECT imaging property for the guidance of radiotherapy. The simply constructed “one-for-all” nanoplatform with radiosensitization and SPECT/MRI imaging function showed great potentials for clinical imaging-guided radiotherapy. Thus, it is highly recommended for publication in Nature Communication. The authors may consider the following suggestions for improvement.

1. Fig. 1e and f showed that oxygen was released from Mb@Gd-NTs in around 10 min, showing the rapid oxygen release. While for long circulation in vivo, such a quick oxygen release might not be effective. Please discuss on this.
2. The authors stated that theoretically, myosin can bind oxygen in pulmonary capillaries, which provides Mb@Gd-NTs with sustainable oxygen source (page 10). Despite a reference showing the oxygen binding by free Gd-Tex was provided, the oxygen binding efficiency for nanosized Mb@Gd-NTs may be different.
3. X-ray dose (total dose and fractions) should be clearly indicated in the figure captions for Fig. 2, 4 and 5, which involved the radiotherapy.
4. The authors are suggested to provide an illustration showing the detailed mechanism of action for Mb@Gd-NTs mediated radiosensitization in vivo (for Fig. 4 and 5).
5. The authors are suggested to provide a timetable for tumor treatment (Fig. 4 and 5) showing when how Mb@Gd-NTs was administered, when X-ray was applied after Mb@Gd-NTs injection, etc.
6. The possible clearance pathways of Cy5.5-Mb and Cy5.5-Mb@Gd-NTs can be discussed based on ex vivo images of organs (Fig. 3g).
7. The radiosensitization mechanism for nimorazole should be discussed.
8. Fig. 5d showed that all Lewis lung cancer-bearing C57 black mouse survived after 120 days after radiotherapy plus Mb@Gd-NTs (20 $\mu\text{mol}/\text{kg}$) treatment. Could this long-term tumor inhibition be related to the antitumor immunological memory formed during treatment? More immunity-related assays could be conducted.
9. The authors may consider citing other references related to nano-radiotherapy in the introduction (Angew. Chem. Inter. Ed., 2021, 60, 12682-12705; Chemical Society Reviews, 2020, 49, 4234-4253.).

Reviewer #2 (Remarks to the Author):

Gd-Tex was a promising radiosensitizer that have been evaluated in clinical trials, but temporarily failed due to insufficient radiosensitization. The authors modified Gd-Tex with lipid to obtain Tex-lipid, and established a versatile liposomal nanoplatform to encapsulate myoglobin for radiosensitization. Now, introducing substances capable of generating oxygen, including hemoglobin/myoglobin, perfluorocarbons, etc., for radiosensitization has been widely reported. In this study, the combination of High-Z strategy and myoglobin were introduced into liposomal nanoparticle for dual radiosensitization. Meanwhile, the experiments are well designed with solid data supporting and the conclusions are convincing. There are some concerns should be addressed before further consideration.

1. The current scheme in Fig. 1a showed the in vitro fabrication processes, which should be more detailed. The mechanism of radiotherapy in vivo should be included in Fig. 1a.
2. Due to the dual Radio-sensitization mechanism, tumor cells and normal tissues would be affected. Although the body weight and blood routine examination preliminarily demonstrated the biosafety of Mb@Gd-NTs + RT, the side effects upon different normal tissues, including intestinal stem cells, should also be detected.
3. The authors compared the radiosensitization efficacy and side effects of Mb@Gd-NTs with nimorazole at different doses under RT. The increased dosage of nimorazole did not further enhanced treatment efficacy and overall survival time. These results are interesting, which should be explained. Why didn't the increased dosage of nimorazole further improve its treatment efficacy?
4. Immunohistochemical (IHC) staining is usually more accuracy than immunofluorescence (IF) staining. Please further perform IHC staining for the experiment of IF staining of γ -H2A.X in Fig. 4E.
5. The authors propose the use of dialysis bags with a molecular cutoff of 1000 kDa to remove unencapsulated Mb. However, the efficiency of dialysis to clear unencapsulated Mb is unknown, and further information is required.
6. The calculation method of "predicted additive effect" in Fig. 2G and 4B should be included in the figure legends or the section of "Methods".
7. The description of statistical tests and sample size should be included in the figure legends, such as Fig. 2H, 3H, 4B-C, 5H-I.
8. The statistical analysis of Fig.5A and 5C should be added.
9. In Fig. 5A and 5C, mg/kg was used as the dose unit of nimorazole, but μ mol/kg was used for Mb@Gd-Tex. To be more intuitive, the same dose unit should be used.

10. All the figure legends do not state whether any of the experiments have been repeated, and whether the results match. This information is essential, especially in cell assays.

11. The combination of High-Z strategy and hypoxia relief for Radiosensitization has been reported (DOI: 10.1002/adfm.201909285), which affects the innovation of this study to a certain extent, and the author needs to further strengthen the discussion on the innovation of this study.

REVIEWER COMMENTS

**Reviewer #1** (Remarks to the Author):

This study reported a lipid nanovesicles (Mb@Gd-NTs) assembled by the gadolinium-
coordinated texaphyrin-lipid (Gd-*Tex*-lipid) that encapsulate oxygen-binding myosin
for radiosensitization. The nanovesicles possessed high loading efficiency of Gd-*Tex*
and long half-life in blood circulation, contributing to the significantly increased Gd-
*Tex* accumulation in tumor compared with free Gd-*Tex*. The oxygen association of
myosin endowed dynamic release of oxygen, improving the tumor oxygenation for the
benefit of radiotherapy. *Tex* was also chelated with radioisotopes (lutetium, ¹⁷⁷Lu) that
endowed Gd-NTs with SPECT imaging property for the guidance of radiotherapy. The
simply constructed “one-for-all” nanoplatfrom with radiosensitization and
SPECT/MRI imaging function showed great potentials for clinical imaging-guided
radiotherapy. Thus, it is highly recommended for publication in Nature Communication.
The authors may consider the following suggestions for improvement.

**Question 1.** Fig. 1e and f showed that oxygen was released from Mb@Gd-NTs in
around 10 min, showing the rapid oxygen release. While for long circulation *in vivo*,
such a quick oxygen release might not be effective. Please discuss on this.

**Response 1.** Thanks for the valuable comments. For *in vitro* assay, oxygenated
21 Mb@Gd-NTs (MbO₂@Gd-NTs) was employed to supply oxygen for hypoxic cells,
which was fabricated by saturating Mb@Gd-NTs with oxygen flow (5 L/min) for 10
23 min. However, for *in vivo* animal experiments, mice were injected with Mb@Gd-NTs
rather than MbO₂@Gd-NTs, because Mb@Gd-NTs could transport arterial oxygen into
deep tumor tissues. Mb@Gd-NTs first absorb oxygen in lung arteries, yielding
MbO₂@Gd-NTs. During its circulation from lung arteries to tumor perfusion arteries,
the release of O₂ from MbO₂@Gd-NTs was quite slow due to the high O₂ concentration
in arterial blood. Therefore, MbO₂@Gd-NTs still contained high concentration of O₂
when reaching tumor perfusion arteries. The oxygen concentration in deep tumor
tissues is significantly lower than tumor perfusion arteries due to the hypoxic tumor
microenvironment. Therefore, MbO₂@Gd-NTs released its O₂ when penetrating deep
tumor tissues from perfusion arteries. Since the penetration process (MbO₂@Gd-NTs
penetrated deep tumor tissues from perfusion arteries) could be completed in seconds,
10 min was enough time to allow the steady release of O₂ and the relief of tumor
hypoxia. The discussion has been added into the revised version of the manuscript.

**Question 2.** The authors stated that theoretically, myosin can bind oxygen in pulmonary
capillaries, which provides Mb@Gd-NTs with sustainable oxygen source (page 10).
Despite a reference showing the oxygen binding by free Gd-Tex was provided, the
oxygen binding efficiency for nanosized Mb@Gd-NTs may be different.

**Response 2.** Thanks for the valuable comments. The oxygen binding efficiency of
nanosized Mb@Gd-NTs was detailedly studied *in vitro* and *in vivo*. To examine oxygen
delivery efficiency *in vitro*, oxygen-saturated Mb@Gd-NTs (MbO₂@Gd-NTs) were
injected into oxygen-free PBS, and the released oxygen was measured using a dissolved
oxygen meter. Mb@Gd-NTs could effectively absorb and gradually release oxygen (Fig.
1F). The oxygen delivery efficiency of Mb@Gd-NTs was similar to that of free Mb
(Fig. 1G), demonstrating that the fabrication procedures of Mb@Gd-NTs did not harm
the bioactivity of Mb.

The oxygen delivery efficiency of nanosized Mb@Gd-NTs was also examined by
cellular experiments *in vitro*. It was investigated whether Mb@Gd-NTs could relieve
hypoxia of tumor cells. Lewis' lung cancer (LLC) cells were maintained at 0.1% O₂ to
induce hypoxic status, followed by incubation with free MbO₂, Gd-NTs, Mb@Gd-NTs,
or MbO₂@Gd-NTs, respectively. Both Gd-NTs and Mb@Gd-NTs barely influenced the
hypoxia status (Fig. 2A, Supplementary Fig. 6A), resulting from the absence of MbO₂.
In contrast, free MbO₂ and MbO₂@Gd-NTs could effectively and equally relieve
cellular hypoxia, demonstrating that nanosized Mb@Gd-NTs could deliver O₂ to
hypoxic cells and the lipid bilayer did not hamper the free diffusion of O₂.

We further investigated whether Mb@Gd-NTs could effectively deliver O₂ to
alleviate tumor hypoxia *in vivo*. Gd-NTs could barely influence tumor hypoxia status,
but Mb@Gd-NTs significantly relieve tumor hypoxia at 24 h post injection (Fig. 4A
and fig. S9), demonstrating Mb in nanosized Mb@Gd-NTs could effectively deliver O₂
to alleviate tumor hypoxia *in vivo*. However, free Mb could hardly influence tumor
hypoxia status (Fig. 3J-K), resulting from its short half-life in blood (Fig. 3I,
Supplementary Table 1). Pegylated stealth nanosized Mb@Gd-NTs could efficiently
prolong the *in vivo* half-life of Mb, increase tumor accumulation, and finally improve
the efficacy of relieving tumor hypoxia. Therefore, nanosized Mb@Gd-NTs has great
advantage on delivering O₂ and relieving hypoxia compared with free Mb.

**Question 3.** X-ray dose (total dose and fractions) should be clearly indicated in the
figure captions for Fig. 2, 4 and 5, which involved the radiotherapy.

**Response 3.** Thanks for the valuable comments. The manuscript has been revised and
X-ray doses (total doses and fractions) have been clearly indicated in the figure captions
of Fig. 2, 4 and 5.

**Question 4.** The authors are suggested to provide an illustration showing the detailed
mechanism of action for Mb@Gd-NTs mediated radiosensitization *in vivo* (for Fig. 4
and 5).

**Response 4.** Thanks for the valuable comments. We have added illustrations for Fig. 4
and Fig. 5. The illustration of Fig. 1A has been modified and *in vivo* functions of
80 Mb@Gd-NTs have been added. A new illustration of Fig. 4H has been added to show
the detailed sensitization mechanism of Mb@Gd-NTs, and a new illustration of Fig. 5A
has been added to show the sensitization mechanism of nimorazole.

**Newly added Fig. 1B.** (A) Diagram of the fabrication of Mb@¹⁷⁷Lu/Gd-NTs. (B)
Illustration of imaging-guided radiosensitization therapy. Intravenous injection of
86 Mb@¹⁷⁷Lu/Gd-NTs enables SPECT/MRI dual-modality imaging for accurately
monitoring drug delivery in real-time. Mb@¹⁷⁷Lu/Gd-NTs can significantly increase
Gd-TeX accumulation in tumor and obviously relieve tumor hypoxia. O₂ synergically
enhances the radiosensitization effect of Gd-TeX, leading to the decreased reductants
and increased ROS in tumor cells, causing the enhanced cell apoptosis, eventually
inducing long-term immunological antitumor memory.

**Newly added Fig. 4H.** The summarized illustration of radiosensitization mechanism of
 94 Mb@Gd-NTs by “futile redox cycle”. RT can generate ROS by the radiolysis of water
 (1), leading to the temporary damage of DNA (2), which can be repaired by reductants
 (3). Gd-Tex can continuously oxidize reductants and generate ROS. The decrease of
 reductants hampers the ability of tumor cells to scavenge ROS (4) and repair DNA
 damage (5), while the increased ROS generated by Gd-Tex can aggregate DNA damage
 (6), leading to the fixed DNA damage and cell death (7). Gd-Tex can also damage other
 biomacromolecules such as proteins and lipids by similar mechanisms (8, 9).

**Question 5.** The authors are suggested to provide a timetable for tumor treatment (Fig.
 4 and 5) showing when how Mb@Gd-NTs was administered, when X-ray was applied
 after Mb@Gd-NTs injection, etc.

**Response 5.** Thanks for the valuable comments. We have provided a timetable for all
 the *in vivo* therapeutic studies, including Fig 4A, 5B, 6A, 7A.

**Newly added Fig. 4A.** Scheme and grouping of *in vivo* therapy (n = 5). The C57BL/6
 mice were inoculated with LLC cells (2×10^6 cells/mouse) on day -9, allowing a tumor
 volume of $\sim 100 \text{ mm}^3$ on day 0. The tumor sites were irradiated with a dose of 2 Gy on
 112 days 0, 2, 4, 6, and 8 for a total dose of 10 Gy. Free Mb was intravenously injected 2 h
 before each RT session, whereas texaphyrin-contained agents were intravenously
 injected 24 h before each RT session with a texaphyrin dose of $5 \mu\text{mol/kg}$ (6.79 mg/kg).

**Newly added Fig. 5B.** Scheme and grouping of *in vivo* therapy (n = 5). The C57BL/6
 mice were inoculated with LLC cells (2×10^6 cells/mouse) on day -9. The tumor sites
 were irradiated with a dose of 2 Gy on days 0, 2, 4, 6, 8, and 10 for a total dose of 12
 120 Gy. Mb@Gd-NTs were intravenously injected with different doses of 6.79-27.15 mg/kg
 24 h before each RT session, whereas nimorazole were i.p. injected with different doses
 of 125-750 mg/kg 30 min before each RT session.

**Newly added Fig. 6A.** Scheme and grouping of *in vivo* therapy. The breast pad of
 BALB/c nude mouse was inoculated with 2×10^6 luciferase-transfected MCF-7 cells
 (MCF-7^{luc+}) on day -7. The tumor site was irradiated with a dose of 2 Gy on day 0, 2,
 4, 6, 8, and 10 for a total dose of 12 Gy. Mb@Gd-NTs were intravenously injected with
 a dose of 27.15 mg/kg 24 h before each RT session, whereas nimorazole were i.p.
 injected with different doses of 125-750 mg/kg 30 min before each RT session.
 Bioluminescence imaging was performed on days 0, 10, and 21 using the IVIS
 Spectrum *in vivo* imaging system.

**Newly added Fig. 7A.** Scheme and grouping of the analysis of long-term immune
 memory. The C57BL/6 mice were inoculated with LLC cells (2×10^6 cells/mouse) on
 137 days -9. For G3 (Mb@Gd-NTs + RT), the tumor sites were irradiated with a dose of 2
 138 Gy on days 0, 2, 4, 6, 8, and 10 for a total dose of 12 Gy, and Mb@Gd-NTs were
 139 intravenously injected with a dose of 27.15 mg/kg (20 μ mol/kg) 24 h before each RT
 session. For G2 (PBS + RT), tumors of mice were irradiated with a dose of 10 Gy on
 141 days 0, 2, 4, 6, and 8 for a total dose of 50 Gy, enabling the eradication of tumors.
 Healthy mice without tumor inoculation (naïve mice) were adopted as controls (G1).

Immune memory cells were analyzed, and the mice were re-challenged with LLC or
MC-38 cells (2×10^6 cells/mouse, subcutaneous inoculation) at day 90.

**Question 6.** The possible clearance pathways of Cy5.5-Mb and Cy5.5-Mb@Gd-NTs
can be discussed based on *ex vivo* images of organs (Fig. 3g).

**Response 6.** Thanks for the valuable comments. The *ex vivo* fluorescence imaging of
major organs showed that livers and spleens had the highest uptake of Cy5.5-Mb@Gd-
NTs compared with other organs, because nanoparticles are easily phagocytosed by the
reticuloendothelial system (RES) that is abundant in livers and spleens. However,
kidneys had the highest uptake of Cy5.5-Mb compared with other organs, and the
accumulation in spleens and lungs was obviously lower than that of Cy5.5-Mb@Gd-
NTs due to the relatively small molecular weight of Mb. The phagocytosed Cy5.5-
155 Mb@Gd-NTs in livers could be metabolized and excreted into bile. Therefore, upon the
156 injection of Cy5.5-Mb@Gd-NTs, the Cy5.5 accumulation in small intestines and large
intestines was higher than the injection of Cy5.5-Mb. In summary, Cy5.5-Mb@Gd-NTs
and Cy5.5-Mb exhibited distinct and typical clearance pathways of nanoparticles and
small molecular proteins. These discussions have been added the revised version of the
manuscript.

*Revised text in the manuscript:*

The biodistribution was also investigated by *ex vivo* fluorescence imaging of
tumors and normal organs performed at 2 h and 24 h after injection of free Cy5.5-Mb
and Cy5.5-Mb@Gd-NTs to compare their maximum uptake (Fig. 3G). Livers and
spleens had the highest uptake of Cy5.5-Mb@Gd-NTs compared with other organs,
because nanoparticles are easily phagocytosed by the reticuloendothelial system (RES)
which is abundant in livers and spleens. However, kidneys had the highest uptake of
Cy5.5-Mb compared with other organs, and the accumulation in spleens and lungs was
obviously lower than that of Cy5.5-Mb@Gd-NTs due to the relatively small molecular
weight of Mb. The phagocytosed Cy5.5-Mb@Gd-NTs in livers could be metabolized
and excreted into bile, leading to higher the Cy5.5 accumulation in small intestines and
large intestines compared with Cy5.5-Mb. In summary, Cy5.5-Mb@Gd-NTs and
Cy5.5-Mb exhibited distinct and typical clearance pathways of nanoparticles and small
molecular proteins.

**Question 7.** The radiosensitization mechanism for nimorazole should be discussed.

**Response 7.** Thanks for the valuable comments. A new illustration has been added for
illustrating the radiosensitization mechanism of nimorazole (Fig. 5A). Radiotherapy is
a treatment modality which uses high-energy photon radiation, such as X-rays and
gamma (γ) rays, and particle radiation, including particles such as alpha (α) or beta (β)
particles, carbon ions, electron (e), proton, or neutron beams. The radiotherapeutic
effects are derived from the direct and indirect destructive action of radiation.
Radiotherapy can directly damage biomolecules (e.g., proteins, lipids, and particularly
DNA) by the high-energy rays or particles, and indirectly damage biomolecules by free
radicals and reactive oxygen (ROS) generated from the radiolysis of water, such as
hydroxyl radical, hydrogen radical, superoxide anion, hydrogen peroxide. The unpaired
electrons of free radicals and ROS induce lesions of biomolecules by chemical reactions
(e.g., hydrogen extraction, addition, disproportionation, and electron capture). These
lesions can be repaired by reductants, such as glutathione (GSH) that contain thiols,
which is an electron-donating group that can neutralize free radicals in injured
biomolecules. Without the reparation of reductants, the lesions in biomolecules lead to
their structural damage, such as single-strand breaks (SSBs) or double-strand breaks
(DSBs) of DNA and cross-linking of DNA-DNA or DNA-protein, eventually resulting
in termination of cell division and proliferation, and even cell necrosis or apoptosis.
Oxygen, which has two unpaired electrons and is a prototype of radiosensitizers, can
rapidly add to many other free radicals, consolidating the lesion of biomolecules,
producing new reactive radicals, and boosting the chain reaction of free radicals.
Nimorazole is a radiosensitizer as the mimic of oxygen, which contains a nitro group
that have electron affinity. The nitro group can rapidly add to free radicals of injured
biomolecules, fixing and amplifying the lesions of biomolecules in a manner similar to
oxygen.

*Revised text in the manuscript:*

**Comparing radiosensitization efficacy of Mb@Gd-NTs with nimorazole**

Nimorazole is a water soluble, 5-nitroimidazole compound that has been widely used
in clinic for the RT of head and neck cancers in Denmark since 1990 (26, 27). We have
compared the radiosensitization efficacy and side effects of Mb@Gd-NTs with the
clinical used nimorazole. The radiosensitization mechanism of nimorazole was
illustrated in Fig. 5A using hydroxyl radical (\bullet OH) as an example. The radiotherapeutic
efficacy greatly relies on the damage of biomolecules (particularly DNA) by free
radicals and ROS, which were generated from the radiolysis of water, including

hydroxyl radical ($\bullet\text{OH}$), hydrogen radical, superoxide anion. The unpaired electrons of
 these free radicals and ROS induce lesions of biomolecules, for example, pyrimidines
 of DNA can react with $\bullet\text{OH}$ to form a carbon radical in pyrimidines (Fig. 5A). These
 lesions can be repaired by reductants, such as GSH. Without the reparation, the lesions
 in biomolecules lead to their structural damage, such as SSBs or DSBs of DNA and
 cross-linking of DNA to DNA, or DNA to proteins, eventually resulting in cell death.
 Oxygen is a prototype of radiosensitizers, and its two unpaired electrons can rapidly
 add to the free radicals in injured biomolecules, consolidating the lesion and boosting
 the chain reaction of free radicals. Nimorazole is a radiosensitizer as the mimic of
 oxygen, which contains a nitro group that have electron affinity. The nitro group can
 also rapidly add to the free radicals of injured biomolecules, consolidating the lesion in
 a manner similar to oxygen.

**Newly added Fig. 5A.** Illustration of the radiosensitization mechanism of nimorazole
 (Ar-NO₂) using hydroxyl radical ($\bullet\text{OH}$) as an example.

**Question 8.** Fig. 5d showed that all Lewis lung cancer-bearing C57 black mouse
 survived after 120 days after radiotherapy plus Mb@Gd-NTs (20 $\mu\text{mol}/\text{kg}$) treatment.
 Could this long-term tumor inhibition be related to the antitumor immunological
 memory formed during treatment? More immunity-related assays could be conducted.

**Response 8.** Thanks for the valuable comments. We have performed a series of
 immunological assays to study the long-term immune memory induced by Mb@Gd-
 NTs + RT. We have found that Mb@Gd-NTs + RT could significantly elicit long-term,
 antigen-specific, and CD8⁺ T cell-mediated immune memory *in vivo* to inhibit tumor
 recurrence. A new section and a new main figure (Fig. 7) have been added into the
 revised version of the manuscript as shown below.

**The long-term immune memory *in vivo* induced by Mb@Gd-NTs + RT**

242 Mb@Gd-NTs + RT could eradicate LLC tumors (Fig. 5C), leading to the long survival
 of 100% mice without tumor recurrence (Fig. 5D). Therefore, we examined whether
 244 Mb@Gd-NTs + RT could elicit long-term immune memory *in vivo* to inhibit tumor

recurrence. As shown in the scheme (**Fig. 7A**), for analyzing the long-term immune
memory, C57BL/6 mice were inoculated with LLC cells on days -9 to allow allowing
a tumor volume of $\sim 100 \text{ mm}^3$ on day 0, which were randomized into two groups: G2,
PBS + RT; G3, Mb@Gd-NTs + RT. Healthy mice without tumor inoculation (naïve
mice) were adopted as controls (G1). Tumors of G2 mice were irradiated with a dose
of 10 Gy on days 0, 2, 4, 6, and 8 for a total dose of 50 Gy to enable the eradication of
tumors. Tumors of G3 mice were irradiated with a dose of 2 Gy on days 0, 2, 4, 6, 8,
and 10 a total dose of 12 Gy, and Mb@Gd-NTs were i.v. injected with a dose of 27.15
253 mg/kg (20 $\mu\text{mol/kg}$) 24 h before each RT session, which also enabling the eradication
of tumors. Immune memory cells were analyzed on days 90. The splenocytes were
collected for analyzing their specific cytotoxicity on homogeneous LLC cells and
heterogenous MC-38 cells. The splenocytes of G3 had specific and significant
cytotoxicity against LLC cells, but showed little influence on the viability of MC-38
cells (**Fig. 7B-C**), demonstrating the antigen specificity of immune response.
Noteworthy, the splenocytes of G3 showed significantly higher cytotoxicity on LLC
cells compared with the splenocytes of G1 and G2 (**Fig. 7B-C**), demonstrating
261 Mb@Gd-NTs + RT generated obvious immune memory, which was stronger than PBS
+ RT. To further prove it, the antigen-specific T cells in splenocytes was quantified by
both flow cytometry (**Fig. 7D-E**) and enzyme-linked immunospot (ELISPOT) assay
(**Fig. 7F**). We found there were more IFN- γ^+ cytotoxic T lymphocytes in splenocytes of
G3, which was 2.13 and 4.79 times as high as that of G2 and G1 with significant
differences ($***P < 0.0001$, **Fig 7E**). Furthermore, G3 elicited significant more central
memory T cell (T_{cm}) and effector memory T cells (T_{em}) both in splenocytes (**Fig. 7G-**
**H**) and blood (**Fig. 7I-J**) compared with G1 and G2. For example, the T_{cm} and T_{em} in
blood of G2 only increased by 64.6% and 52.9% compared with G1, but the T_{cm} and
T_{em} in blood of G3 dramatically increased by 201.9% 144.1% compared with G1 (**Fig.**
**7J**). Therefore, Mb@Gd-NTs + RT could generate obvious antigen-specific CD8^+ T
cell-mediated immune memory, which was much stronger than PBS + RT.

To further study the long-term immune memory, tumor-free mice of G1, G2, and
G3 (obtained on days 90) were re-challenged with LLC or MC-38 cells (subcutaneous
inoculated with these tumor cells), and tumor growth was recorded for 20 days. G3
protected 100% mice from the rechallenge of LLC cells, while G2 only protected 33.3%
mice from the rechallenge of LLC cells (**Fig. 7K**), further demonstrating the more
powerful immune memory induced by Mb@Gd-NTs + RT compared with PBS + RT.
Meanwhile, both G2 and G3 could hardly inhibit the growth of MC-38 cells, further

proving the antigen-specificity of immune memory.

Newly added Fig. 7. The long-term immune memory *in vivo* induced by Mb@Gd-

NTs + RT. (A) Scheme and grouping of the analysis of long-term immune memory.

The C57BL/6 mice were inoculated with LLC cells (2×10^6 cells/mouse) on days -9.

For G3 (Mb@Gd-NTs + RT), the tumor sites were irradiated with a dose of 2 Gy on

days 0, 2, 4, 6, 8, and 10 for a total dose of 12 Gy, and Mb@Gd-NTs were intravenously

injected with a dose of 27.15 mg/kg (20 μ mol/kg) 24 h before each RT session. For G2

(PBS + RT), tumors of mice were irradiated with a dose of 10 Gy on days 0, 2, 4, 6, and

8 for a total dose of 50 Gy, enabling the eradication of tumors. Healthy mice without

tumor inoculation (naïve mice) were adopted as controls (G1). Immune memory cells

were analyzed, and the mice were re-challenged with LLC or MC-38 cells (2×10^6

cells/mouse, subcutaneous inoculation) at day 90. (B-C) Cell viability of LLC (B) and
MC-38 cells (C) after the incubation with splenocytes collected at day 90 examined by
LDH release assay, demonstrating the specific killing ability of splenocytes. (D-E)
Flow cytometry analysis of IFN- γ ⁺ cytotoxic T lymphocytes in splenocytes re-
stimulated with whole cell lysate of LLC cells. (F) IFN- γ secretion by the splenocytes,
as determined by ELISPOT assay after re-stimulation with whole cell lysate of LLC
cells. (G-J) Quantitative analysis of effector memory T cells (T_{em}, CD3⁺CD8⁺CD62L⁻
CD44⁺), central memory T cells (T_{cm}, CD3⁺CD8⁺CD62L⁺CD44⁺) and *naive* T cells
(T_{naive}, CD3⁺CD8⁺CD62L⁺CD44⁻) in splenocytes (G-H) and blood (I-J), showing the
radiosensitization-elicited T cell memory. (K-L) Tumor growth curves of LLC (K) and
MC-38 (L). Tumor-free mice (n = 6 mice) of G1, G2, and G3 (obtained at day 90) were
re-challenged with LLC or MC-38 cells, and the tumor growth was recorded for 20
305 days. The data (B-C, E-F, H, and J) are shown as the mean \pm SD (n = 5 mice). Statistical
analysis was performed by a two-tailed unpaired *t* test. *, *P* < 0.05; **, *P* < 0.01; ***, *P* <
0.001.

**Question 9.** The authors may consider citing other references related to nano-
radiotherapy in the introduction (Angew. Chem. Inter. Ed., 2021, 60, 12682-12705;
Chemical Society Reviews, 2020, 49, 4234-4253.).

**Response 9.** Thanks for the valuable comments. We have added the references related
to nano-radiotherapy in the introduction.

**Reviewer #2** (Remarks to the Author):

Gd-Tex was a promising radiosensitizer that have been evaluated in clinical trials, but
temporarily failed due to insufficient radiosensitization. The authors modified Gd-Tex
with lipid to obtain Tex-lipid, and established a versatile liposomal nanoplatform to
encapsulate myoglobin for radiosensitization. Now, introducing substances capable of
generating oxygen, including hemoglobin/myoglobin, perfluorocarbons, etc., for
radiosensitization has been widely reported. In this study, the combination of High-Z
strategy and myoglobin were introduced into liposomal nanoparticle for dual
radiosensitization. Meanwhile, the experiments are well designed with solid data
supporting and the conclusions are convincing. There are some concerns should be
addressed before further consideration.

**Question 1.** The current scheme in Fig. 1a showed the in vitro fabrication processes,

which should be more detailed. The mechanism of radiotherapy *in vivo* should be
included in Fig. 1a.

**Response 1.** Thanks for the valuable comments. The mechanism of radiosensitization
of Mb@Gd-NTs *in vivo* has been included in the newly added Fig. 1B. Besides, we
have added a new illustration (Fig. 4H) to further describe the mechanism of
radiosensitization of Mb@Gd-NTs *in vivo* at cellular level in more detail. The
mechanism of radiosensitization of nimorazole has also been illustrated in the Fig. 5A
according to the comment of Reviewer 1# (Question 7#).

**Newly added Fig. 1B.** (A) Diagram of the fabrication of Mb@¹⁷⁷Lu/Gd-NTs. (B)
Illustration of imaging-guided radiosensitization therapy. Intravenous injection of
339 Mb@¹⁷⁷Lu/Gd-NTs enables SPECT/MRI dual-modality imaging for accurately
monitoring drug delivery in real-time. Mb@¹⁷⁷Lu/Gd-NTs can significantly increase
Gd-Tex accumulation in tumor and obviously relieve tumor hypoxia. O₂ synergically
enhances the radiosensitization effect of Gd-Tex, leading to the decreased reductants
and increased ROS in tumor cells, causing the enhanced cell apoptosis, eventually
inducing long-term immunological antitumor memory.

**Newly added Fig. 4H.** The summarized illustration of radiosensitization mechanism of
 347 Mb@Gd-NTs by “futile redox cycle”. RT can generate ROS by the radiolysis of water
 (1), leading to the temporary damage of DNA (2), which can be repaired by reductants
 (3). Gd-Tex can continuously oxidize reductants and generate ROS. The decrease of
 reductants hampers the ability of tumor cells to scavenge ROS (4) and repair DNA
 damage (5), while the increased ROS generated by Gd-Tex can aggregate DNA damage
 (6), leading to the fixed DNA damage and cell death (7). Gd-Tex can also damage other
 biomacromolecules such as proteins and lipids by similar mechanisms (8, 9).

**Question 2.** Due to the dual radio-sensitization mechanism, tumor cells and normal
 tissues would be affected. Although the body weight and blood routine examination
 preliminarily demonstrated the biosafety of Mb@Gd-NTs + RT, the side effects upon
 different normal tissues, including intestinal stem cells, should also be detected.

**Response 2.** Thanks for the valuable comments. Gd-Tex + RT is quite safe with
 minimal side effects as proved by the phase II/III clinical trials. The radiosensitization
 of Gd-Tex relies on the reduce of intracellular reductants (i.e., GSH, NADPH) and the
 generation of intracellular ROS. Since tumor cells have a higher level of intracellular
 ROS compared with normal cells, Gd-Tex + RT is more cytotoxic to tumor cells than
 normal cells. Besides, Mb@Gd-NTs could specifically accumulate in tumors owing to
 the enhanced permeability and retention (EPR) effect, as shown by the SPECT/MRI
 (Fig. 3C-D), fluorescence imaging (Fig. 3F-G), and biodistribution study (Fig. 3E).
 367 Mb@Gd-NTs + RT could further reduce the side effects of Gd-Tex due to its specific
 accumulation in tumors and the high T/NT (target/non-target) ratios of drug
 concentration.

Radiation injure of intestinal stem cells leads to radiation enteritis, which is the

most common side effect of intestinal radiotherapy. Besides, radiation pneumonitis is
the most common side effect of pulmonary radiotherapy. Therefore, we have examined
radiation enteritis and pneumonitis to investigate the radiation-induced side effects on
lungs and intestines. C57BL/6 mice were divided into three groups with different
treatments including PBS (G1'), PBS + RT (G2'), Mb@Gd-NTs + RT (G3'). Mouse
abdomen or chest was respectively irradiated with a dose of 2 Gy for 6 times as shown
in the schedule of **Supplementary Fig. 15A**. Mb@Gd-NTs were i.v. injected with a
dose of 27.15 mg/kg (the highest therapeutic dose, Fig. 5B-D) 24 h before each RT
session. Oxidation stress and inflammatory responses are typical symptoms of
radiation-induced pneumonitis and enteritis. To examining the oxidation stress and
inflammatory responses of lungs and intestines, the reactive oxygen species (ROS),
myeloperoxidase (MPO), and interleukin-1 β (IL-1 β) were analyzed at two weeks after
the last radiation. Mb@Gd-NTs + RT had little influence on ROS, MPO, and IL-1 β of
lungs (**Fig. 5F**) and intestines (**Fig. 5G**). Therefore, Mb@Gd-NTs + RT could hardly
induce radiation pneumonitis and enteritis. Radiation fibrosis is one of the most serious
side effects of pulmonary and intestinal radiotherapy. Masson staining of tissue section
can directly show the percentage and distribution of collagen fibers. Mb@Gd-NTs +
RT also had little influence on the collagen fibers of lungs (**Fig. 5H, Supplementary**
**Fig. 15B-C**), colorectum (**Fig. 5I, Supplementary Fig. 16A-B**), and small intestines
(**Supplementary Fig. 16C-E**). Therefore, Mb@Gd-NTs + RT could hardly induce
radiation fibrosis, showing its high biosafety.

Radiation toxicity of healthy tissues can be reduced by fractionated low-dose
radiation compared with single high-dose radiation, although longtime and repetitive
radiation of 50 Gy in 25 fractions can also induce inflammation and subsequent fibrosis
(*Science Translational Medicine* 2018, 10, eaan0333). Mouse models of radiation-
induced intestinal injury are usually established by irradiating mice with a single high
dose (*Nature Communication* 2014, 5:3492). In this manuscript, an exposure planning
of fractionated low-dose radiation was employed, which could hardly cause radiation
enteritis and pneumonitis. Besides, stereotactic body radiation therapy (SBRT) is
widely used in clinic, which applies stereotactic technique and special ray generating
equipment that focuses high energy rays of multi-source, multi-beam on a target
(converge rays from different directions to a single point), leading to high dose of the
target and low dose of normal tissues (*Lancet oncology* 2023, 24(3), e121-e132).
Therefore, if tumors were irradiated with a dose of 2 Gy for 6 times, SBRT could enable
a much less radiation dose on lungs and intestines, which could hardly induce radiation

injures.

**Newly added Fig. 5F-I.** The examination of radiation-induced side effects of intestines
 and lungs. C57BL/6 mice were treated with PBS (G1'), PBS + RT (G2'), and Mb@Gd-
 NTs + RT (G3') as shown in the schedule of Supplementary Fig. 15A, whereas mouse
 abdomen or chest was irradiated instead of tumors with a dose of 2 Gy on days 0, 2, 4,
 6, 8, 10 for a total dose of 12 Gy. The oxidation stress and inflammatory responses of
 the lung (H) and intestine (I) was analyzed on days 25. Masson staining of lung (F) and
 colorectum (G) sections showed the distribution of collagen (blue) in tissues. Scale bar
 of the right panels, 100 μ m. The data (F-G) are shown as the mean \pm SD (n = 5 mice).
 Statistical analysis was performed by a two-tailed unpaired *t* test. *, *P* < 0.05; **, *P* <
 0.01; ***, *P* < 0.001.

**Newly added Supplementary Figure 16.** The examination of radiation-induced
 fibrosis of lungs. (A) The schedule of examining radiation-induced side effects of

intestine and lung. 6-week-old C57BL/6 mice were divided into three groups (n =
 5) with different treatments: (G1') PBS; (G2') PBS + RT; (G3') Mb@Gd-NTs + RT.
 For evaluating side effects on intestine or lung, mouse abdomen or chest was
 respectively irradiated with a dose of 2 Gy on days 0, 2, 4, 6, 8, 10 for a total dose of
 12 Gy. Mb@Gd-NTs were i.v. injected with a dose of 27.15 mg/kg (20 μ mol/kg) 24 h
 before each RT session. Mice were sacrificed for necropsy on days 25, and the intestines
 and lungs were collected for the analysis of oxidation stress and inflammatory
 responses and Masson staining of tissue sections. (B-C) Masson staining of lung
 sections of G1' (B) and G2' (C) showing the distribution of collagen in lungs (n = 5
 mice). Cells and nucleus were stained red and dark blue with ponceau and hematoxylin,
 while collagen fibers were stained blue with aniline blue. Scale bar for the right panels,
 100 μ m. (D) Semi-quantification of mean integrated optical density (IOD) of fiber in
 lung sections. Mean IOD = (IOD of fiber-positive region) / (Total area of tissue). The
 data are shown as the mean \pm SD (n = 5 mice). Statistical analysis was performed by a
 two-tailed unpaired *t* test. *, *P* < 0.05; **, *P* < 0.01; ***, *P* < 0.001.

 **Newly added Supplementary Figure 17. The examination of radiation-induced**
 **fibrosis of intestines.** (A-B) Masson staining of colorectum sections of G1' (A) and
 G2' (B) showing the distribution of collagen in colorectum (n = 5 mice). (C-E) Masson
 staining of small intestine sections of G1' (C), G2' (D), and G3' (E) showing the
 distribution of collagen in small intestines (n = 5 mice). Cells and nucleus were stained
 red and dark blue with ponceau and hematoxylin, while collagen fibers were stained

blue with aniline blue. Scale bar for the enlarged views, 100 μm . (F) Semi-
quantification of mean integrated optical density (IOD) of fibers in large and small
intestine sections. The data are shown as the mean \pm SD (n = 5 mice). Statistical analysis
was performed by a two-tailed unpaired *t* test. *, $P < 0.05$; **, $P < 0.01$; ***, $P < 0.001$.

**Question 3.** The authors compared the radiosensitization efficacy and side effects of
449 Mb@Gd-NTs with nimorazole at different doses under RT. The increased dosage of
450 nimorazole did not further enhance treatment efficacy and overall survival time. These
451 results are interesting, which should be explained. Why didn't the increased dosage of
452 nimorazole further improve its treatment efficacy?

**Response 3.** Thanks for the valuable comments. Nimorazole is a water soluble, 5-
nitroimidazole compound that has been widely used in clinic for the radiotherapy of
head and neck cancers in Denmark since 1990. The radiosensitization mechanism of
nimorazole was illustrated in **Fig. 5A**. The radiotherapeutic efficacy greatly relies on
the damage of biomolecules (particularly DNA) by free radicals and ROS generated
from the radiolysis of water, including hydroxyl radical ($\bullet\text{OH}$), hydrogen radical,
superoxide anion. The unpaired electrons of these free radicals and ROS induce lesions
of biomolecules. These lesions can be repaired by reductants, such as GSH. Without
the reparation of reductants, the lesions in biomolecules lead to their structural damage,
such as SSBs or DSBs of DNA and cross-linking of DNA-DNA or DNA-protein,
eventually resulting in cell death. Oxygen is a prototype of radiosensitizers, and its two
unpaired electrons can rapidly add to the free radicals in injured biomolecules,
consolidating the lesion and boosting the chain reaction of free radicals. Nimorazole is
a radiosensitizer as the mimic of oxygen, which contains a nitro group that have electron
affinity. The nitro group can also rapidly add to the free radicals of injured biomolecules,
consolidating the lesion in a similar manner to oxygen.

A literature has reported the unusual lack of dose-response relationship over a wide
dose range of nimorazole (*British Journal of Cancer* 1982, 46, 904). Nimorazole has
the "maximum" radiosensitization effect, and the effect cannot be elevated by
increasing its dose. The literature has compared the radiosensitization effect of
nimorazole with misonidazole (MISO) in the same animal tumor model. For a single-
dose irradiation, nimorazole gives an enhancement ratio (ER) of ~ 1.4 , independent of
the dose of drug administered over the range 0.1-10 mg/g (Table II of the literature).
MISO yields a similar ER at the 0.1 mg/g level. But, unlike nimorazole, MISO shows
a steep dose-response curve with an ER of 2.2 when given in a concentration of 1.0

478 mg/g (Table II of the literature).

The difference of nimorazole and MISO in dose-response relationship is not
 resulted from their pharmacokinetics, because tumor and plasma concentrations of the
 two drugs have an identical dose relationship (Fig. 2 in the literature). The tumor
 concentration of nimorazole can significantly increase with an increasing dose.
 Although the lack of dose-response relationship has been reported, its mechanism has
 not been demonstrated until now.

TABLE II.—Comparison of the effect of MISO and nimorazole on the response of single-dose radiation

Dose of sensitizer ^a (mg/g)	MISO		Nimorazole		ER MISO ER nimorazole
	TCD ₅₀ (Gy)	ER	TCD ₅₀ (Gy)	ER	
None (radiation alone)	56.2 (54.5-57.9)	—	56.2 (54.5-57.9)	—	—
1.0	25.7 (23.5-28.1)	2.18 (2.03-2.35)	37.6 (33.3-42.5)	1.49 (1.40-1.59)	1.46 (1.18-1.81)
0.8	—	—	38.2 (33.0-44.1)	1.47 (1.32-1.63)	—
0.5	34.2 (30.4-38.2)	1.65 (1.50-1.80)	39.0 (35.4-42.9)	1.44 (1.34-1.55)	1.15 (0.89-1.48)
0.3	35.9 (31.8-39.3)	1.56 (1.46-1.67)	39.7 (34.4-46.6)	1.42 (1.31-1.53)	1.10 (0.87-1.38)
0.1	40.0 (37.0-43.2)	1.41 (1.32-1.50)	41.4 (37.3-46.1)	1.36 (1.24-1.48)	1.04 (0.82-1.32)

^a Given 30 min before radiation.
 Numbers in parentheses represent 95% confidence limits.

FIG. 2.—Relationship between given drug dose and concentrations in plasma and tumour measured 30 min after i.p. injection. MISO: tumour/plasma ratio=0.51; ○, plasma slope=0.825, $r=0.9888$; ●, tumour slope=0.418, $r=0.9778$. Nimorazole: tumour/plasma ratio=0.50; □, plasma slope=0.769, $r=0.9981$; ■, tumour slope=0.384, $r=0.9915$.

Table II and Fig. 2 of a previous literature showing the unusual lack of dose-response
 relationship over a wide dose range of nimorazole (*British Journal of Cancer* 1982, 46,
 904).

**Question 4.** Immunohistochemical (IHC) staining is usually more accuracy than
immunofluorescence (IF) staining. Please further perform IHC staining for the
experiment of IF staining of γ -H2A.X in Fig. 4E.

**Response 4.** Thanks for the valuable comments. We have performed IHC staining of γ -
H2A.X according to the advice. The IHC staining of γ -H2A.X exhibited similar results
to the IF staining. LLC tumors were given with X-ray radiation, and different agents
including PBS, Gd-NTs, Mb@NTs, and Mb@Gd-NTs were *i.v.* injected 24 h before
each radiation. The Gd-NTs + RT and Mb@NTs + RT induced significant more DNA
double-strand breaks (DSBs) compared with PBS + RT, demonstrating the obvious
radiosensitization effect of Gd-NTs and Mb@NTs, which respectively resulted from the
radiosensitization of Gd-Tex and the relief of tumor hypoxia. Mb@Gd-NTs + RT
induced the highest level of DSBs compared with Gd-NTs + RT and Mb@NTs + RT,
demonstrating the synergistic effect of Gd-Tex and O₂. These results have been added
to the revised version of the manuscript.

Newly added Fig. 4G. γ -H2A.X immunohistochemical staining of tumor tissue sections. Scale bar: 180 μ m (upper panel) and 30 μ m (bottom panel).

**Supplementary Figure 11B.** Semi-quantification of mean integrated optical density
(IOD) of γ -H2A.X in tumor tissue sections of Figure 4G. Mean IOD = (IOD of γ -
H2A.X-positive region) / (Total area of tissue).

**Question 5.** The authors propose the use of dialysis bags with a molecular cutoff of
1000 kDa to remove unencapsulated Mb. However, the efficiency of dialysis to clear
unencapsulated Mb is unknown, and further information is required.

**Response 5.** Thanks for the valuable comments. The molecular weight of Mb is ~ 17
517 kDa, which is much lower than the molecular cutoff of dialysis bag (1000 kDa).
Theoretically, the unencapsulated Mb can be almost completely removed by the dialysis
bag. We have confirmed the efficiency of dialysis using size exclusion chromatography
(SEC). SEC separates different molecules based on their size by filtration through a gel.
The gel consists of spherical beads containing pores of a specific size distribution.
Separation occurs when molecules of different sizes are included or excluded from the
pores within the matrix. Small molecules (e.g., free Mb) diffuse into the pores and their
flow through the column is retarded according to their size, while large molecules do
not enter the pores and are eluted in the column's void volume. Consequently, molecules
separate based on their size as they pass through the column and are eluted in order of
decreasing molecular weight (MW).

The unencapsulated Mb (globular proteins) and Mb@Gd-NTs can be separated
using the gel exclusion column containing Sephadex G-100, which has a fractionation
range of 4000-150000 Da for globular proteins. Free Mb or Mb@Gd-NTs was added to
the gel exclusion columns, and the protein concentration of eluent was measured using
BCA (bicinchoninic acid) Protein Assay Kit by analyzing the absorption at 562 nm.
Gd-NTs had little interference to the BCA assay of Mb concentration, because Gd-Tex-
Lipid had a low absorption at 562 nm (Fig. 1E). As shown in the following figure, free
535 Mb was eluted in volume fractions between 6.3 and 9.2 ml with the peak at 7.6 mL, but
536 Mb@Gd-NTs (purified by dialysis) was quickly eluted in volume fractions between 5.5
and 6.9 ml with the peak at 6.2 mL, because the large size (~ 110 nm) of Mb@Gd-NTs
did not allow itself to enter the pores of Sephadex G-100, leading to the quick elution
in the column's void volume. Mb@Gd-NTs (purified by dialysis) showed only one peak
in the chromatogram, demonstrating that the unencapsulated Mb can be almost
completely removed by the dialysis bag. These results and discussions have been added
to the revised version of the manuscript.

**Supplementary Figure 2. Examination of unencapsulated Mb in Mb@Gd-NTs by**
 **size exclusion chromatography.** The possible unencapsulated Mb (globular protein, ~
 17 kDa) in Mb@Gd-NTs was examined using the gel exclusion column containing
 Sephadex G-100 with a fractionation range of 4000-150000 Da for globular proteins.
 Free Mb or Mb@Gd-NTs was added to the columns, and the protein concentration of
 eluent was measured using BCA (bicinchoninic acid) Protein Assay Kit. Free Mb was
 eluted in volume fractions between 6.3 and 9.2 ml with the peak at 7.6 mL, but
 551 Mb@Gd-NTs (purified by dialysis) was quickly eluted in volume fractions between 5.5
 and 6.9 ml with the peak at 6.2 mL, which showed only one peak in the chromatogram,
 demonstrating that the unencapsulated Mb can be almost completely removed by the
 dialysis bag with a molecular cutoff of 1000 kDa.

**Question 6.** The calculation method of “predicted additive effect” in Fig. 2G and 4B
 should be included in the figure legends or the section of “Methods”.

**Response 6.** Thanks for the valuable comments. The necessary information and
 reference have been added to the figure legend of Fig. 2F-G and the section of
 “Methods”.

**Fig. 2F-G.** *In vitro* radiosensitization effect examined by colony formation assay.
 Photograph of colonies (F) and survival fraction (G) of LLC cells. Cells were treated
 with different agents under hypoxic conditions, and irradiated with X-rays at total doses
 of 2, 4, 6, or 8 Gy for one time. Scale bar, 2 cm. “Predicted additive effect” means the
 calculated theoretical additive radiosensitization effect of Gd-NTs and MbO₂@NTs
 (assuming there was no synergistic effect), which was calculated by multiplying the
 survival inhibition ratios of Gd-NTs + RT and MbO₂@NTs + RT according to the
 reported method (42).

**Question 7.** The description of statistical tests and sample size should be included in
the figure legends, such as Fig. 2H, 3H, 4B-C, 5H-I.

**Response 7.** Thanks for the valuable comments. The description of statistical tests and
sample sizes have been included in the figure legends of Fig. 2H, 3H, 4B-C, 5H-I.

**Question 8.** The statistical analysis of Fig.5A and 5C should be added.

**Response 8.** Thanks for the valuable comments. The statistical analysis has been added
in the revised version of the manuscript.

**Newly added Supplementary Table 2.** Statistical analysis of Fig. 5D. Statistical
analysis was performed by a two-tailed unpaired *t* test. *, $P < 0.05$; **, $P < 0.01$; ***, $P <$
**0.001.**

	G1	G2	G3	G4	G5	G6	G7	G8
G2	* $P =$ 0.0244							
G3	** $P =$ 0.0018	** $P =$ 0.0064						
G4	** $P =$ 0.0018	** $P =$ 0.0018	** $P =$ 0.0018					
G5	** $P =$ 0.0018	** $P =$ 0.0018	** $P =$ 0.0018	$P =$ 0.1642				
G6	** $P =$ 0.0018	** $P =$ 0.0018	** $P =$ 0.0018	$P =$ 0.0830	$P =$ 0.9505			
G7	** $P =$ 0.0018	** $P =$ 0.0018	** $P =$ 0.0018	$P =$ 0.5739	$P =$ 0.2917	$P =$ 0.1546		
G8	** $P =$ 0.0018	** $P =$ 0.0018	** $P =$ 0.0018	** $P =$ 0.0018	** $P =$ 0.0064	** $P =$ 0.0064	** $P =$ 0.0018	
G9	** $P =$ 0.0018	** $P =$ 0.0018	** $P =$ 0.0018	** $P =$ 0.0018	** $P =$ 0.0018	** $P =$ 0.0018	** $P =$ 0.0018	* $P =$ 0.0494

**Question 9.** In Fig. 5A and 5C, mg/kg was used as the dose unit of nimorazole, but
$\mu\text{mol/kg}$ was used for Mb@Gd-Tex. To be more intuitive, the same dose unit should be
used.

**Response 9.** Thanks for the valuable comments. The dose unit of Mb@Gd-Tex has
been changed to mg/kg in the revised version of the manuscript.

**Question 10.** All the figure legends do not state whether any of the experiments have
been repeated, and whether the results match. This information is essential, especially
in cell assays.

**Response 10.** Thanks for the valuable comments. We have thoroughly checked and
revised all the figure legends to ensure that the suggested information has been added.

**Question 11.** The combination of high-Z strategy and hypoxia relief for
radiosensitization has been reported (DOI: 10.1002/adfm.201909285), which affects
the innovation of this study to a certain extent, and the author needs to further strengthen
the discussion on the innovation of this study.

**Response 10.** Thanks for the valuable comments. We have strengthened the discussion
on the innovation of this study accordingly. The main purpose of this study is to improve
the radiosensitization of gadolinium (Gd^{3+})-coordinated texaphyrin (Gd-*Tex*), but not
to develop a new kind of radiosensitizer such as high-Z materials. The corresponding
commercial drug of Gd-*Tex* was Motexafin gadolinium (Xcytrin[®]) developed by
Pharmacyclics[®] in the United States, which had undergone lots of clinical trials.
Although Gd-*Tex* was eventually not approved by the US FDA due to limited
radiosensitization efficacy, the accomplishment of its phase III clinical trials proved
that it had good safety. If the radiosensitization efficacy of Gd-*Tex* could be effectively
improved, it is hopeful to achieve its successful clinical application, benefiting
numerous patients receiving radiotherapy.

Our study had innovations in the following aspects:

(1) Little attention has been given to using nanovesicles to improve the
radiosensitization efficacy of Gd-*Tex*. In this study, Gd-*Tex* was transformed into
building blocks “Gd-*Tex*-lipids” to self-assemble nanovesicles, realizing high density
packing of Gd-*Tex* in a single nanovesicle, and achieving significantly prolonged
circulation in blood and largely increased Gd-*Tex* accumulation in tumor to enhance
radiosensitization.

(2) Little attention has been given to the O₂ dependence of the radiosensitization
of Gd-*Tex*. However, hypoxia is the hallmark of most solid tumors, and therefore it is
essential to elucidate the O₂ dependence of Gd-*Tex* radiosensitization. Herein, a
synergistic effect by appropriate O₂ delivery and enhanced radiosensitization of Gd-*Tex*
has been demonstrated by the in-depth studies. The nanovesicles enabled
spatiotemporal codelivery of O₂ and high density of Gd-*Tex* into tumors, resulting in
efficient relief of tumor hypoxia and the significant enhancement of Gd-*Tex*
radiosensitization. The O₂ dependence of Gd-*Tex* radiosensitization is possibly one of
the main reasons of its failure in clinical trials.

(3) The “one-for-all” nanovesicles owed the capability of chelating various

trivalent metal ions. The chelation of $^{177}\text{Lu}^{3+}$ and Gd^{3+} enabled SPECT/MRI dual-
modality imaging-guided radiosensitization therapy. The SPECT/MRI dual-modality
imaging can not only clearly show the anatomical location of tumors, but can also
quantitatively display the drug biodistribution in real time, which is helpful to
investigate the *in vivo* behavior of pharmacokinetics and distribution, and facilitates to
determine the administration time of external beam irradiation.

Besides, Gd-Tex and its nanovesicle Mb@Gd-NTs do not belong to high-Z
materials, owing to the special chemical formula and radiosensitization mechanism,
which is different from traditional high-Z materials. Most of high Z nanomaterials are
inorganic metal nanomaterials such as gold (Au) and silver (Ag) nanoparticles, bismuth
(Bi)-based nanomaterials (Bi_2S_3 , Bi_2Se_3 , Cu_3BiS_3 and Bi_2O_3), Gd-based nanomaterials
($\text{GdW}_{10}\text{O}_{36}$), tungsten (W)-based nanomaterials (WS_2 , SnWO_4 , WO_{3-x}), hafnium (Hf)-
based nanomaterials (HfO_2), tantalum oxide (TaO_x) nanoshells (*Advanced Materials*,
2018, 1802244). Many of these high Z nanomaterials are difficult to be degraded *in*
*in vivo* and have long-term safety hazards. However, the small molecular Gd-Tex is
organic coordination compounds, which has excellent degradability, and its good
biosafety has been confirmed by lots of clinical trials. Mb@Gd-NTs is liposome
composed of Gd-Tex-Lipid, phospholipids, cholesterol, and myoglobin, all of which
have good biosafety and can be degraded *in vivo*. The high biocompatibility and simple
construction exhibit broad clinical application prospects of Mb@Gd-NTs.

650 Mb@Gd-NTs also has special radiosensitization mechanism, which is different
from traditional high-Z materials. The radiosensitization of Mb@Gd-NTs relies on the
“futile redox cycle” as demonstrated in the manuscript (Fig. 4H). As illustrated in the
following figure, the radiosensitization of high Z nanomaterials relies on their strong
capabilities for absorbing irradiation and emitting secondary electrons, leading to local
dose enhancement, mainly via Compton scattering or the photoelectric effect. Compton
scattering, an inelastic scattering of a photon by a charged particle (usually an electron),
is one of the most important interactions in radiotherapy. It results in a decrease in
energy of the incident photon by collision with electrons and hence can deposit the
irradiation energy at the local site. For the photoelectric effect, electrons within a certain
atom shell absorb the energy of incident irradiation and are ejected to ionize
surrounding biomolecules. These effects, along with others, such as the production of
Auger electrons and electron pairs as well as coherent scattering, enable high Z
nanomaterials to increase the irradiation dose in tumor site. Therefore, the
radiosensitization mechanism of Gd-Tex and Mb@Gd-NTs is quite different from

traditional high Z nanomaterials.

Figure 4 in the published literature (*Trends in Pharmacological Sciences*, 2018, 39, 24-
48) demonstrating the mechanisms of the high-Z materials in radiosensitization. High-
Z nanomaterials (gold nanoparticles, for example) can interact with ionizing radiation
(IR) and biomolecules in several ways. The nanomaterials deposit the IR dose through
interactions, such as the Compton effect, photoelectric effect, and Auger effect, leading
to DNA damage via direct and indirect effects.

REVIEWERS' COMMENTS

Reviewer #1 (Remarks to the Author):

The authors have made their effort to conduct additional experiments to fully address the reviewers' questions and concerns. The relevant parts in the main text and experiment sections are also properly revised. Thus, it is recommended for publication.

Reviewer #2 (Remarks to the Author):

The authors have addressed the concerns. I recommend accepting this manuscript.

**REVIEWERS' COMMENTS**

**Reviewer #1** (Remarks to the Author):

The authors have made their effort to conduct additional experiments to fully address the reviewers'
questions and concerns. The relevant parts in the main text and experiment sections are also properly
revised. Thus, it is recommended for publication.

**Response:** Thank you for your recognition of our work. Thank you for your great effort and
constructive suggestions for our manuscript.

**Reviewer #2** (Remarks to the Author):

The authors have addressed the concerns. I recommend accepting this manuscript.

**Response:** Thank you for your recognition of our work. Thank you for your great effort and
constructive suggestions for our manuscript.